# A Curriculum Perspective of Robust Loss Functions

## Abstract

Learning with noisy labels is a fundamental problem in machine learning. A large body of work aims to design loss functions robust against label noise. However, it remain open questions why robust loss functions can underfit and why loss functions deviating from theoretical robustness conditions can appear robust. To tackle these questions, we show that a broad array of loss functions differs only in the implicit sample-weighting curriculums they induce. We then adopt the resulting curriculum perspective to analyze how robust losses interact with various training dynamics, which helps elucidate the above questions. Based on our findings, we propose simple fixes to make robust losses that severely underfit competitive to state-of-the-art losses. Notably, our novel curriculum perspective complements the common theoretical approaches focusing on bounding the risk minimizers.[1]

## 1 Introduction

Labeling errors are non-negligible [1] in datasets from expert annotation [2, 3], crowd-sourcing [4] and automatic annotation [5, 6]. The resulting noisy labels can hamper generalization, as over-parameterized neural networks can memorize all training samples [7]. To combat the impact of noisy labels, a large body of research aims to design loss functions robust against label noise [8–13]. Theoretical results show that loss functions satisfying certain robustness conditions [9, 11] will lead to the same optimum with clean or noisy labels.

Existing approaches focus on bounding the risk minimizer of a loss function [9–11, 14, 15] with the presence of label noise, which are agnostic to the training dynamics. Though theoretically appealing, they may fail to fully characterize the performance of robust losses with noisy labels. In particular, it has been shown that (1) robust losses can underfit difficult tasks [1, 10, 12, 13], while (2) losses failing to satisfy theoretical robustness conditions [12, 13, 16] can exhibit robustness. The reasons behind these observations remain open questions. For (1), existing explanations [10, 17] can be limited as discussed in §2.3. For (2), to our knowledge, there is no work directly addressing it.

To tackle the above questions, we consider training dynamics in our analysis, which complements existing theoretical approaches [9–11]. By rewriting loss functions into a standard form, we show that many loss function differs in the implicitly sample-weighting curriculums they induce (§3), which connects robust losses to the seemingly distinct [1] curriculum learning approaches [18–22] for noise-robust training. The original definition [23] of curriculum learning aims to present training samples with gradually increasing difficulty and diversity to ease learning. We adopt a generalized definition of curriculum [24], i.e., a *curriculum* specifies a sequence of *re-weighting* of training sample distributions, which can manifest as sample weighting [18–20] or sample selection [21, 22, 25].

The curriculum perspective helps elucidate underfitting and noise robustness from the interaction between the sample-weighting curriculums and various training dynamics. We first attribute un-derfitting to the marginal average sample weights with the implicit curriculums (§4.1). We then show that an increased number of classes can lead to marginal *initial* sample weights with some loss

---

[1]Our code will be available at `github`.

functions (§4.2). By adapting their curriculums accordingly, we make robust losses that severely underfit perform competitively to state-of-the-art loss functions (§4.2). Finally, we attribute the noise robustness of loss functions to higher average sample weights for clean samples compared to noisy ones (§4.3). We hypothesize that clean samples can receive higher weights with sample-weighting curriculums that magnify the learning speed differences and neglect unlearnt samples, which explains our empirical observations (§4.3). Inspired by this hypothesis, we find two unexpected results when viewed from existing theoretical robustness guarantees: by simply changing the learning rate schedule, robust losses can be vulnerable to label noise and cross entropy can appear robust (§4.3).

## 2  Background

After formulating classification with label noise, we briefly review typical sufficient conditions and loss functions for noise robustness to set the context for our novel curriculum perspective. We then summarizing open questions to be addressed in this work.

### 2.1  Classification with Label Noise and Noise Robustness

The $k$-ary classification problem with input $\boldsymbol{x} \in \mathbb{R}^d$ can be solved with classifier $\arg\max_i s_i$, where $s_i$ is the score of the $i$-th class from the class scoring function $\boldsymbol{s}_\theta : \mathbb{R}^d \to \mathbb{R}^k$. The class scores $\boldsymbol{s}_\theta(\boldsymbol{x})$ can be turned into class probabilities with the softmax function $p_i = e^{s_i}/(\sum_{j=1}^k e^{s_j})$, where $p_i$ is the probability for class $i$. Given a loss function $L(\boldsymbol{s}_\theta(\boldsymbol{x}), y)$ and data $(\boldsymbol{x}, y)$ with ground truth label $y \in \{1, \ldots, k\}$, the parameter $\theta$ of $\boldsymbol{s}_\theta$ can be estimated with risk minimization $\arg\min_\theta \mathbb{E}_{\boldsymbol{x},y} L(\boldsymbol{s}_\theta(\boldsymbol{x}), y)$, whose solution are called risk minimizers. For notation simplicity, we omit the dependence on $\theta$ and $\boldsymbol{x}$ if possible.

The annotation process may introduce errors, resulting in a potentially corrupted label $\tilde{y}$ following

$$\tilde{y} = \begin{cases} y, & \text{with probability } P(\tilde{y}=y|\boldsymbol{x},y) \\ i, i \neq y & \text{with probability } P(\tilde{y}=i|\boldsymbol{x},y) \end{cases}$$

Label noise is symmetric (or uniform) if $P(\tilde{y}=i|\boldsymbol{x},y) = \eta/(k-1), \forall i \neq y$, with $\eta = P(\tilde{y} \neq y)$ the noise rate constant. Label noise is asymmetric (or class-conditional) if $P(\tilde{y}=i|\boldsymbol{x},y) = P(\tilde{y}=i|y)$. Given data $(\boldsymbol{x}, \tilde{y})$ with noisy label $\tilde{y}$, a loss function $L$ is robust against label noise if

$$\arg\min_\theta \mathbb{E}_{\boldsymbol{x},\tilde{y}} L(\boldsymbol{s}_\theta(\boldsymbol{x}), \tilde{y}) = \arg\min_\theta \mathbb{E}_{\boldsymbol{x},y} L(\boldsymbol{s}_\theta(\boldsymbol{x}), y) \tag{1}$$

Most existing work [9–11, 14, 15] aim to derive bounds for the difference between risk minimizers obtained using noisy and clean data, i.e., ensuring Eq. (1) holds with some conditions. As typical examples, loss functions satisfying the symmetric [9] or asymmetric [11] conditions are theoretically guaranteed to be robust. A loss function $L$ is called *symmetric* if

$$\sum_i L(\boldsymbol{s}_\theta(\boldsymbol{x}), i) = C, \forall \boldsymbol{x}, \boldsymbol{s}_\theta \tag{2}$$

where $C$ is a constant. When noise rate $\eta < (k-1)/k$, a symmetric loss is robust against symmetric label noise [9]. Such stringent condition is later relaxed by Zhou et al. [11]. Suppose a loss function can be written as a function of softmax probability $p_i$, i.e., $L(\boldsymbol{s}_\theta(\boldsymbol{x}), i) = l(p_i)$. As an equivalent rephrase of the sufficient condition, $L$ is called *asymmetric* if

$$\max_{i \neq y} \frac{P(\tilde{y}=i|\boldsymbol{x},y)}{P(\tilde{y}=y|\boldsymbol{x},y)} = \tilde{r} \leq r = \inf_{\substack{0 \leq p_i, \Delta p \leq 1 \\ p_i + \Delta p \leq 1}} \frac{l(p_i) - l(p_i + \Delta p)}{l(0) - l(\Delta p)} \tag{3}$$

where $\Delta p$ is a valid increment of $p_i$. When clean labels dominate the data, i.e., $\tilde{r} < 1$, an asymmetric loss is robust against *generic* label noise. Notably, both symmetric and asymmetric conditions for noise robustness are agnostic to training dynamics to reach the risk minimizers.

### 2.2  Review of Selected Loss Functions

In addition to cross entropy (CE) that is vulnerable to label noise [9], we review typical loss functions for later analysis. We *ignore differences in constant scaling factors and constant additive bias* in the loss functions. They are either equivalent to learning rate scaling in SGD or irrelevant in the gradient computation. See Table 1 for the exact formulas and Appendix A for an extended review.

| Type | Name | Function | Sample Weight $w$ | Constraints |
|------|------|----------|-------------------|-------------|
| | CE | $-\log p_y$ | $1 - p_y$ | |
| Sym. | MAE/RCE | $1 - p_y$ | $p_y(1 - p_y)$ | |
| | NCE | $\frac{-\log p_y}{\sum_{i=1}^{k} -\log p_i}$ | $\gamma_{\mathrm{NCE}}\left(w_{\mathrm{CE}} + \frac{k-1}{k}\epsilon_{\mathrm{NCE}}\right)$ | |
| Asym. | AUL | $\frac{(a - p_y)^q - (a-1)^q}{q}$ | $p_y(1 - p_y)(a - p_y)^{q-1}$ | $a > 1, q > 0$ |
| | AGCE | $\frac{(a+1) - (a + p_y)^q}{q}$ | $p_y(a + p_y)^{q-1}(1 - p_y)$ | $a > 0, q > 0$ |
| Comb. | GCE | $\frac{1 - p_y^q}{q}$ | $p_y^q(1 - p_y)$ | $0 < q \leq 1$ |
| | SCE | $(1-q) \cdot L_{\mathrm{CE}} + q \cdot L_{\mathrm{MAE}}$ | $(1 - q + q \cdot p_y)(1 - p_y)$ | $0 < q < 1$ |
| | NCE+MAE | $(1-q) \cdot L_{\mathrm{NCE}} + q \cdot L_{\mathrm{MAE}}$ | $(1-q) \cdot w_{\mathrm{NCE}} + q \cdot w_{\mathrm{MAE}}$ | $0 < q < 1$ |

Table 1: Expressions, constraints of hyperparameters and sample weights of the implicit curriculums (§3.1) for loss functions reviewed in §2.2. Note that $w_{\mathrm{NCE}}$ is an approximation as discussed in §3.2.

**Symmetric Loss** The mean absolute error (MAE) [9] and the subsequent reverse cross entropy (RCE) [13] are essentially equivalent, both satisfying Eq. (2). Ma et al. [10] normalize generic loss functions satisfying $L(\boldsymbol{s}, i) > 0, \forall i \in \{1, \ldots, K\}$ into symmetric losses with $L_{\mathrm{N}}(\boldsymbol{s}, y) = L(\boldsymbol{s}, y) / (\sum_{i=1}^{k} L(\boldsymbol{s}, i))$. We include normalized cross entropy (NCE) as an example.

**Asymmetric Loss** We include two asymmetric losses [11] for our analysis: asymmetric generalized cross entropy (AGCE) and asymmetric unhinged loss (AUL). Notably, AGCE with $q \geq 1$ and AUL with $q \leq 1$ are both completely asymmetric [11], i.e., Eq. (3) always holds when $\tilde{r} < 1$.

**Combined Loss** Loss functions can be combined for both robust and sufficient learning. For example, generalized cross entropy (GCE) [12] can be viewed as a smooth interpolation between CE and MAE. Alternatively, symmetric cross entropy (SCE) [13] uses a weighted average of CE and RCE (MAE). Finally, Ma et al. [10] argue that robust and sufficient training requires a balanced combination of active and passive losses. Suppose loss function $L$ can be rewritten into

$$L(\boldsymbol{s}, y) = \sum_{i=1}^{k} l(\boldsymbol{s}, i) \tag{4}$$

where $l$ is a function of scores $\boldsymbol{s}$ and any possible label $i$. An active loss requires $\forall i \neq y, l(\boldsymbol{s}, i) = 0$, which focuses on learning the target label. In contrast, a passive one satisfies $\exists j \neq y, l(\boldsymbol{s}, i) \neq 0$, which can improve by unlearning non-target labels. Accordingly, CE and NCE are active while MAE (RCE) is passive. We use NCE+MAE as an example.

## 2.3 Open Questions

**Why do robust losses underfit?** Ma et al. [10] attribute underfitting to failure in balancing active-passive components. However, different specifications of Eq. (4) can lead to ambiguities in the active-passive dichotomy. For example, with $L_{\mathrm{MAE}}(\boldsymbol{s}, y) \propto \sum_{i=1}^{k} |\mathbb{I}(i = y) - p_y|$ where $\mathbb{I}(\cdot)$ is the indicator function, MAE is passive; yet the equivalent $L_{\mathrm{MAE}}(\boldsymbol{s}, y) \propto \sum_{i=1}^{k} \mathbb{I}(i = y)(1 - p_y)$ makes MAE an active loss. Wang et al. [17] instead view $\|\nabla_{\boldsymbol{s}} L(\boldsymbol{s}, y)\|_1$ as weights for sample gradients and attribute underfitting to their low variance, making clean and noisy samples less distinguishable. However, as we show in §4.1, MAE also underfits on clean datasets. Why robust losses underfit thus remains an open question.

**What affects the robustness of a loss function?** Although combined losses such as GCE and SCE fail to satisfy existing robustness conditions (Eq. (2) and (3)), it is unclear why they exhibit robustness against label noise [12, 13]. Furthermore, it is unclear how training dynamics, which are irrelevant in many theoretical robustness guarantees [9–11, 14, 15], affect the noise robustness of a loss function.

## 3 Implicit Curriculums of Robust Loss Functions

We derive the standard form of reviewed loss functions and show that each implicitly induces a sample-weighting curriculum, which helps examine how they interact with various training dynamics.

### 3.1 The Implicit Sample-Weighting Curriculums

Loss functions in Table 1 are generally functions of the target softmax probability $p_y$, i.e., $L(\boldsymbol{s}, y) = l(p_y)$. Note that $p_y$ can be rewritten as

$$p_y = \frac{e^{s_y}}{\sum_{i=1}^k e^{s_i}} = \frac{1}{e^{\log \sum_{i \neq y} e^{s_i - s_y}} + 1} = \frac{1}{e^{-\Delta_y} + 1} \tag{5}$$

where

$$\Delta_y = s_y - \log \sum_{i \neq y} e^{s_i} \leq s_y - \max_{i \neq y} s_i = \Delta_y^* \tag{6}$$

is the *soft margin* between $s_y$ and the maximum of other scores, a smooth approximation of the *hard margin* $\Delta_y^*$. $\Delta_y$ indicates how well a sample is learnt given classifier $\arg\max_i s_i$, as $\Delta_y \geq 0$ leads to $\Delta_y^* \geq 0$, ensuring successful classification with scores $\boldsymbol{s}$. Since $\nabla_{\boldsymbol{s}} l(p_y) = l'(p_y) \cdot p_y'(\Delta_y) \cdot \nabla_{\boldsymbol{s}} \Delta_y$, these loss functions can be rewritten into a standard form with *equivalent gradients*:

$$L(\boldsymbol{s}, y) = -\operatorname{stop\_grad}[w(\Delta_y)] \cdot \Delta_y \tag{7}$$

where $\operatorname{stop\_grad}(\cdot)$ avoids backpropagating through $w(\Delta_y) = l'(p_y) \cdot p_y'(\Delta_y)$. The equivalence is valid only with first-order derivatives. Each loss function *in the form of* Eq. (7) thus implicitly induces a sample-weighting curriculum, where $w(\Delta_y)$ is the *sample weight* and $\Delta_y$ the *implicit loss*. By examining how $w(\Delta_y)$ interacts with different training dynamics, we can elucidate the reasons behind underfitting and noise robustness. Table 1 summarizes $w(\Delta_y)$ for the reviewed loss functions.

Wang et al. [16, 17] treat $\|\nabla_{\boldsymbol{s}} L(\boldsymbol{s}, y)\|_1$ as weights for sample gradients, which share similar formulas as $w(\Delta_y)$ in Table 1. Instead of directly weighting sample gradients, our derivation identifies the implicit loss $\Delta_y$, making our sample-weighting scheme compatible with the definition of curriculum learning [24]. In addition, the extracted $\Delta_y$ and $\Delta_y^*$ can serve as direct metrics for sample performance in curriculums, compared to loss [26, 27] and gradient magnitude [28] that are affected by preference from $w(\Delta_y)$ of a loss function. Finally, the $\Delta_y$ distribution is essential in analyzing the interaction between loss functions and training dynamics in §4.

### 3.2 The Additional Entropy-Reducing Curriculum of NCE

Due to normalization, $L_{\mathrm{NCE}}(\boldsymbol{s}, y)$ in Table 1 additionally depends on $\Delta_i$ where $i \neq y$, which cannot be be trivially rewritten into Eq. (7). A derivation of the gradient gives

$$\nabla_{\boldsymbol{s}} L_{\mathrm{NCE}}(\boldsymbol{s}, y) = \frac{1}{\sum_{i=1}^k L_{\mathrm{CE}}(\boldsymbol{s}, i)} \left\{ \nabla_{\boldsymbol{s}} L_{\mathrm{CE}}(\boldsymbol{s}, y) + \frac{k L_{\mathrm{CE}}(\boldsymbol{s}, y)}{\sum_{i=1}^k L_{\mathrm{CE}}(\boldsymbol{s}, i)} \cdot \nabla_{\boldsymbol{s}} \left[ -\frac{1}{k} \sum_{i=1}^k L_{\mathrm{CE}}(\boldsymbol{s}, i) \right] \right\}$$
$$= \gamma_{\mathrm{NCE}} \cdot [\nabla_{\boldsymbol{s}} L_{\mathrm{CE}}(\boldsymbol{s}, y) + \epsilon_{\mathrm{NCE}} \cdot \nabla_{\boldsymbol{s}} R_{\mathrm{NCE}}(\boldsymbol{s})]$$

Thus NCE can be rewritten as

$$L_{\mathrm{NCE}}(\boldsymbol{s}, y) = \gamma_{\mathrm{NCE}} \cdot L_{\mathrm{CE}}(\boldsymbol{s}, y) + \gamma_{\mathrm{NCE}} \cdot \epsilon_{\mathrm{NCE}} \cdot R_{\mathrm{NCE}}(\boldsymbol{s}) \tag{8}$$

In this equivalent form, there is no backpropagation through the computation of $\gamma_{\mathrm{NCE}}$ and $\epsilon_{\mathrm{NCE}}$. The first term results in a similar sample-weighting curriculum as CE, with an additional factor $\gamma_{\mathrm{NCE}} = 1/(\sum_{i=1}^k -\log p_i) \leq 1/(k \log k)$. The second term is a regularizer

$$R_{\mathrm{NCE}}(\boldsymbol{s}) = -\frac{1}{k} \sum_{i=1}^k L_{\mathrm{CE}}(\boldsymbol{s}, i) \tag{9}$$

which reduces the entropy of the softmax output. The regularizer has per-sample weights $\epsilon_{\mathrm{NCE}} = k(-\log p_y)/(\sum_{i=1}^k -\log p_i)$. It can thus be interpreted as a regularization curriculum. Notably, the two curriculums work synergically in reducing the entropy of the softmax output.

The extra regularizer makes NCE incompatible to Eq. (7). However, as shown in Appendix C, since $\Delta_y$ induces gradients with constant L1 norm, we can *approximate* the upperbound of $w_{\mathrm{NCE}}$ with

$$w_{\mathrm{NCE}} = \frac{\|\nabla_{\boldsymbol{s}} L_{\mathrm{NCE}}(\boldsymbol{s}, y)\|_1}{\|\nabla_{\boldsymbol{s}} \Delta_y\|_1} \leq \gamma_{\mathrm{NCE}} \left( w_{\mathrm{CE}} + \frac{k-1}{k} \epsilon_{\mathrm{NCE}} \right) \tag{10}$$

See Appendix C for derivations. Note that directions of $\nabla_{\boldsymbol{s}} L_{\mathrm{NCE}}(\boldsymbol{s}, y)$ and $\nabla_{\boldsymbol{s}} \Delta_y$ may be different.

| Underfitting | Loss | CIFAR100 Acc. | $\bar{\alpha}_t^*$ | CIFAR10 Acc. | $\bar{\alpha}_t^*$ |
|---|---|---|---|---|---|
| No | CE | $71.33 \pm 0.23$ | 8.183 | $92.76 \pm 0.30$ | 5.541 |
| | GCE | $69.95 \pm 0.40$ | 8.861 | $92.96 \pm 0.13$ | 6.151 |
| | SCE | $71.36 \pm 0.39$ | 9.541 | $93.17 \pm 0.06$ | 7.018 |
| | NCE+MAE | $68.89 \pm 0.23$ | 2.971 | $92.37 \pm 0.33$ | 2.414 |
| Moderate | NCE | $43.18 \pm 1.55$ | 1.769 | $91.28 \pm 0.22$ | 1.072 |
| | AUL | $58.75 \pm 1.07$ | 5.278 | $92.43 \pm 0.19$ | 5.171 |
| | AGCE | $49.27 \pm 1.03$ | 4.537 | $92.61 \pm 0.18$ | 5.225 |
| Severe | MAE | $3.69 \pm 0.59$ | 0.035 | $91.56 \pm 0.11$ | 2.492 |
| | AUL$^\dagger$ | $3.13 \pm 0.43$ | 0.033 | $91.13 \pm 0.06$ | 2.308 |
| | AGCE$^\dagger$ | $1.62 \pm 0.69$ | 0.009 | $87.14 \pm 4.96$ | 1.701 |

Table 2: With clean labels, robust losses can underfit CIFAR100 but CIFAR10. Hyperparameters of loss functions are tuned on CIFAR100 and listed in Table 7. We report test accuracy and average effective learning rate $\bar{\alpha}_t^*$ (scaled by $10^3$) at the final training step with 3 different runs, using learning rate $\alpha = 0.1$. AUL$^\dagger$ and AGCE$^\dagger$ with inferior hyperparamters are included as reference. See Appendix D for results with $\alpha = 0.01$.

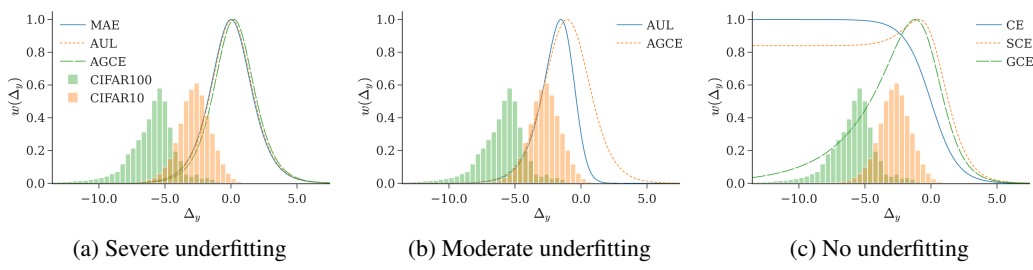

(a) Severe underfitting      (b) Moderate underfitting      (c) No underfitting

Figure 1: Sample-weighting functions $w(\Delta_y)$ for selected loss functions and hyperparameters in Table 2. We include the initial distributions of $\Delta_y$ on CIFAR10 and CIFAR100 for reference.

## 4 Understanding Robust Losses with Their Implicit Curriculums

We empirically investigate the interaction between sample-weighting curriculums and various training dynamics for questions in §2.3. Experiments are conducted on MNIST [29] and CIFAR10/100 [30] with synthetic symmetric and asymmetric label noises following standard settings [10, 11]. We also include real human noisy labels provided by Wei et al. [31] on CIFAR10/100. We use a 4-layer CNN for MNIST, an 8-layer CNN for CIFAR10 and a ResNet-34 [32] for CIFAR100. By default, models are trained with a fixed number of epochs using SGD with momentum, weight decay and cosine learning rate annealing. See Appendix B for more experimental details. Different from standard settings, we rescale $w(\Delta_y)$ to have unit maximum to avoid complications, since hyperparameters of loss functions can change the maximum of $w(\Delta_y)$, essentially adjusting the learning rate of SGD.

### 4.1 Underfitting of Robust Losses from a Sample-Weighting Curriculum Perspective

**Robust losses can underfit.** We confirm that on difficult tasks like CIFAR100 [10, 12, 13], underfitting results from robust losses themselves rather than inferior hyperparameters. We tune hyperparameters of loss functions on CIFAR100 and report results on CIFAR100 and CIFAR10 without label noise. As shown in Table 2, the performance of NCE, AGCE and AUL lag behind CE by a nontrivial margin on CIFAR100. Notably, MAE performs much worse compared to CE, similar to AGCE$^\dagger$ and AUL$^\dagger$ with inferior hyperparameters. In contrast, all loss functions fit CIFAR10 well. See Table 8 in Appendix D for similar results with a smaller learning rate.

**Marginal effective learning rate explains underfitting.** We attribute underfitting to the diminishing effective learning rate $\alpha_t^* = \alpha_t \cdot \bar{w}_t$, where $\bar{w}_t$ is the average sample weight of the batch and $\alpha_t$ the learning rate at step $t$. We use the average effective learning rate up to step $t$, $\bar{\alpha}_t^* = \sum_{i=1}^t \alpha_i^*/t$, to characterize the overall $\alpha_t^*$. In Table 2, for loss functions that heavily underfit on CIFAR100, their $\bar{\alpha}_t^*$ at the final step is marginal compare to CE.

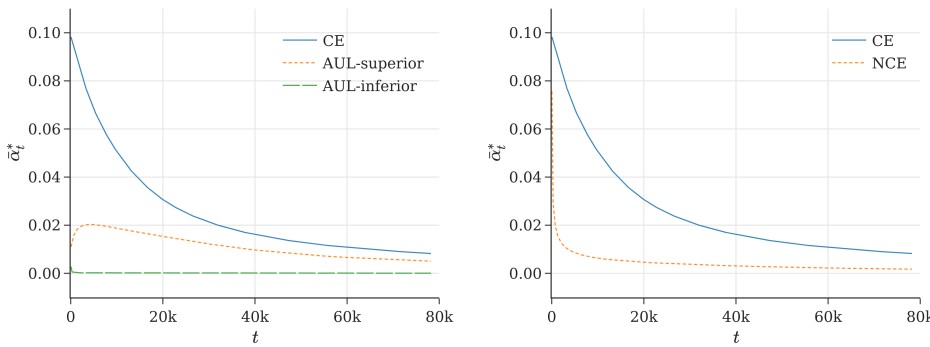

(a) AUL with inferior/superior hyperparameters.     (b) NCE with estimated weight upperbound.

Figure 2: Different causes of underfitting: (a) marginal initial sample weights; (b) fast diminishing sample weights. We plot the average effective learning rate $\bar{\alpha}_t^*$ at different training steps $t$ with selected loss functions on CIFAR100.

**Marginal effective learning rate due to marginal initial sample weights.** In Fig. 1 we compare sample-weighting functions $w(\Delta_y)$ of robust losses to the $\Delta_y$ distribution of CIFAR10 and CIFAR100 at initialization. For robust losses that severely underfit (Fig. 1a), the $\Delta_y$ distribution of CIFAR100 concentrates at regions with marginal sample weights, resulting in small effective learning rate $\alpha_t^*$. It can be hard for these samples to escape the region with marginal weights before the learning rate attenuates. In contrast, loss functions with non-trivial initial sample weights (Fig. 1b and 1c) result in moderate or no underfitting in Table 2. As a corroboration, we plot the average effective learning rate $\bar{\alpha}_t^*$ of AUL with different hyperparameters in Fig. 2a. With superior hyperparameters (AUL in Table 2), $\bar{\alpha}_t^*$ quickly increase to a non-negligible value before annealing. In contrast, $\bar{\alpha}_t^*$ stays marginal with inferior hyperparameters (AUL$^\dagger$ in Table 2).

**Marginal effective learning rate due to fast diminishing sample weights.** In Fig. 2b, different from other robust losses but similar to CE, the effective learning rate of NCE peaks at initialization. However, it decreases much faster compared to CE, which can be attributed to the synergy between the two implicit curriculums of NCE in reducing $w_{\text{NCE}}$. As $\Delta_y$ improves, $\gamma_{\text{NCE}}$, $\epsilon_{\text{NCE}}$ and $w_{\text{CE}}$ all decreases. In addition, the regularizer $R_{\text{NCE}}(\boldsymbol{s})$ further decreases the entropy of softmax output and thus $\gamma_{\text{NCE}}$. Thus $w_{\text{NCE}}$ decreases much faster compared to $w_{\text{CE}}$, leading to faster attenuating $\alpha_t^*$.

**Loss combination mitigates marginal initial sample weights.** As $w_{\text{CE}}$ and $w_{\text{NCE}}$ peak at initialization, they compensate the marginal initial sample weights when combined with other robust losses, helping initial learning and thus avoiding underfitting. In Table 2, the effective learning rate on CIFAR100 is substantially increased when combining MAE with CE and NCE. Interestingly, CE and NCE are both "active" as their sample weights peak at initialization, while other robust losses are "passive" due to their marginal initial sample weights. Such dichotomy based on sample-weighting curriculums complements the active-passive dichotomy [10] from a distinct perspective.

### 4.2 Addressing Underfitting by Adapting the Sample-Weighting Curriculums

As shown in Table 2, robust losses can underfit on CIFAR100 but CIFAR10. Such difference has been vaguely attributed to the increased task difficulty [1, 12]. We further show that with static sample-weighting curriculums, loss functions suffer from *marginal initial sample weights* due to the increased number of classes $k$. By adapting the curriculums accordingly, robust losses that severely underfit can become competitive with the state-of-the-art. We leave the fix for NCE to future work, and use MAE as a typical example for illustration.

Intuitively, the larger number of classes, the more subtle differences to be distinguished, thus the harder the task is. In addition, the number of classes $k$ determines the $\Delta_y$ distribution at initialization. Assuming that class scores $s_i$ at initialization are i.i.d. variables following the normal distribution, i.e., $s_i \sim \mathcal{N}(\mu, \sigma)$. In particular, $\mu = 0$ and $\sigma = 1$ for most neural networks with standard initializations [33] and normalization layers [34, 35]. See Appendix E for comparisons between simulations and real settings. The expected $\Delta_y$ can be approximated with

$$\mathbb{E}[\Delta_y] \approx -\log(k-1) - \sigma^2/2 + \frac{e^{\sigma^2} - 1}{2(k-1)} \tag{11}$$

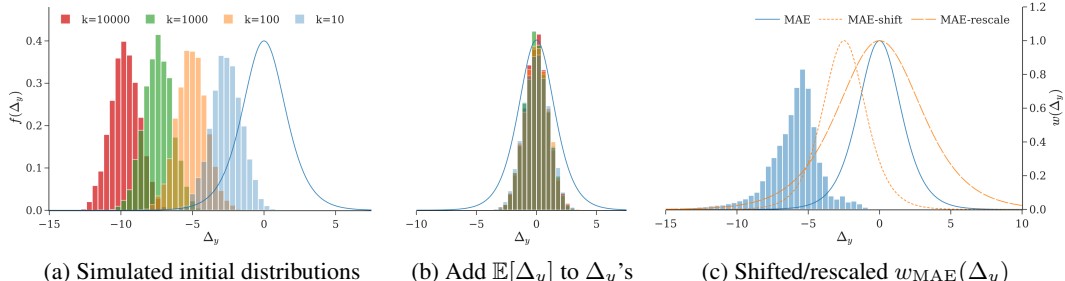

(a) Simulated initial distributions    (b) Add $\mathbb{E}[\Delta_y]$ to $\Delta_y$'s    (c) Shifted/rescaled $w_{\mathrm{MAE}}(\Delta_y)$

Figure 3: (a). Simulated initial $\Delta_y$ distributions with different $k$ assuming $s_i \sim \mathcal{N}(\mu, \sigma)$. We include the plot of $w_{\mathrm{MAE}}(\Delta_y)$ for reference. (b). Adding $\mathbb{E}[\Delta_y]$ to $\Delta_y$'s centers simulated distributions in (a) to the origin. (c). The shifted and rescaled $w_{\mathrm{MAE}}(\Delta_y)$ with $a = 2.6$ and $k = 100$. We include the initial $\Delta_y$ distribution of CIFAR100 for reference.

|  | Clean | Symmetric | | Asymmetric | Human |
|---|---|---|---|---|---|
| Loss | $\eta = 0$ | $\eta = 0.4$ | $\eta = 0.8$ | $\eta = 0.4$ | $\eta = 0.4$ |
| CE [11] | $71.33 \pm 0.43$ | $39.92 \pm 0.10$ | $7.59 \pm 0.20$ | $40.17 \pm 1.31$ | |
| GCE [11] | $63.09 \pm 1.39$ | $56.11 \pm 1.35$ | $17.42 \pm 0.06$ | $40.91 \pm 0.57$ | |
| NCE [11] | $29.96 \pm 0.73$ | $19.54 \pm 0.52$ | $8.55 \pm 0.37$ | $20.64 \pm 0.40$ | |
| NCE+AUL [11] | $68.96 \pm 0.16$ | $59.25 \pm 0.23$ | $23.03 \pm 0.64$ | $38.59 \pm 0.48$ | |
| AGCE | $49.27 \pm 1.03$ | $47.76 \pm 1.75$ | $16.03 \pm 0.59$ | $33.40 \pm 1.57$ | $30.45 \pm 1.50$ |
| AGCE shift | $67.50 \pm 1.48$ | $53.33 \pm 1.08$ | $10.47 \pm 0.57$ | $38.37 \pm 1.55$ | $44.44 \pm 1.39$ |
| AGCE rescale | $67.20 \pm 0.79$ | $56.32 \pm 0.59$ | $12.75 \pm 1.10$ | $40.00 \pm 0.27$ | $49.08 \pm 0.74$ |
| MAE | $3.69 \pm 0.59$ | $1.29 \pm 0.50$ | $1.00 \pm 0.00$ | $2.53 \pm 1.34$ | $2.09 \pm 0.55$ |
| MAE shift | $69.02 \pm 0.78$ | $44.60 \pm 0.24$ | $8.08 \pm 0.26$ | $40.57 \pm 0.47$ | $48.31 \pm 0.31$ |
| MAE rescale | $69.95 \pm 1.21$ | $60.70 \pm 0.30$ | $10.79 \pm 0.97$ | $39.22 \pm 1.54$ | $54.65 \pm 0.73$ |

Table 3: Shifting or rescaling $\Delta_y$ mitigates underfitting on CIFAR100 with different noise types and noise rate $\eta$. Human noisy labels are from CIFAR100-N [31]. Test accuracies are reported with 3 different runs. We use $a = 4.5$ for AGCE and $a = 2.6$ for MAE. Results from [11] are included as context. See Appendix E for results on WebVision and CIFAR100 with additional noise rates.

We leave detailed derivations to Appendix E. With more output classes, the $\Delta_y$ distribution will have smaller expectation, corresponding to diminishing initial sample weights with the fixed MAE curriculum, as shown in Fig. 3a. In Fig. 3b, subtracting $\mathbb{E}(\Delta_y)$ from $\Delta_y$ centers distributions to 0.

**Shifting or rescaling $w(\Delta_y)$ mitigates underfitting from increased number of classes.** To assign nontrivial sample weights at initialization, the sample-weighting curriculum of robust losses should be adapted according to the number of classes $k$. A simple strategy is to make the expected initial sample weights agnostic to $k$. Given a sample-weighting function $w(\Delta_y)$, we can either shift

$$w^{\mathrm{shift}}(\Delta_y) = w(\Delta_y + \mathbb{E}[\Delta_y] - a) \tag{12}$$

or rescale

$$w^{\mathrm{rescale}}(\Delta_y) = w(\Delta_y / \mathbb{E}[\Delta_y] \cdot a) \tag{13}$$

it, where $a > 0$ is a hyperparameter. The shifted and scaled $w_{\mathrm{MAE}}(\Delta_y)$ are shown in Fig. 3c as an illustration. Intuitively, shifting or scaling with $\mathbb{E}[\Delta_y]$ can cancel the effect of increased $k$ on the expected initial sample weights. With smaller $a$, samples will get higher weights at initialization.

In Table 3, we test our fixes with different noise types and noise rates on CIFAR100. See Appendix E for more results on the large scale noisy dataset WebVision [36] and CIFAR100 with different synthetic noise rates. Rescaling and shifting alleviate the underfitting issues, making MAE and AGCE perform comparable to the previous best (NCE+AUL) [11]. Notably, the performance of MAE is substantially improved. Interestingly, despite being effective fixes for underfitting, simply scaling or shifting $w(\Delta_y)$'s can risk assigning large weights for noisy samples, which have lower $\Delta_y$ in general as discuss in §4.3, thus hampering the noise robustness of loss functions. Under symmetric label noise with $\eta = 0.8$, the performance of AGCE decreases after applying the fixes.

| Loss | Clean Acc | Symmetric | | | | | | | | Human | |
|---|---|---|---|---|---|---|---|---|---|---|---|
| | | $\eta = 0.2$ | | $\eta = 0.4$ | | $\eta = 0.6$ | | $\eta = 0.8$ | | $\eta = 0.4$ | |
| | | $\Delta_{\text{acc}}$ | snr | $\Delta_{\text{acc}}$ | snr | $\Delta_{\text{acc}}$ | snr | $\Delta_{\text{acc}}$ | snr | $\Delta_{\text{acc}}$ | snr |
| CE | 90.49 | -15.85 | 0.39 | -32.34 | 0.58 | -51.57 | 0.77 | -71.14 | 0.95 | -28.18 | 0.53 |
| SCE | 91.06 | -8.10 | 0.76 | -21.55 | 1.03 | -43.86 | 1.29 | -71.10 | 1.32 | -22.96 | 0.74 |
| GCE | 90.85 | -2.02 | 3.25 | -5.59 | 3.16 | -14.16 | 2.95 | -50.10 | 2.29 | -12.52 | 1.14 |
| MAE | 90.56 | -1.96 | 3.46 | -8.25 | 3.15 | -12.31 | 2.88 | -38.11 | 2.53 | -22.49 | 1.00 |
| AUL | 90.79 | -1.90 | 3.51 | -5.06 | 3.40 | -13.43 | 3.01 | -50.99 | 1.79 | -22.36 | 1.02 |
| AGCE | 90.56 | -4.28 | 3.11 | -4.47 | 3.29 | -17.76 | 2.69 | -44.87 | 2.04 | -21.62 | 1.02 |

Table 4: Robust losses assign larger weights to clean samples. We report snr and drop in test accuracy with symmetric and human label noise on CIFAR10 at the final step with 3 different runs. We use the "worst" version of CIFAR10-N [31] as human label noise. Standard deviation are omitted due to space limitation. Hyperparameters of loss functions are tuned with noise rate $\eta = 0.6$. See Appendix B for detailed hyperparameters.

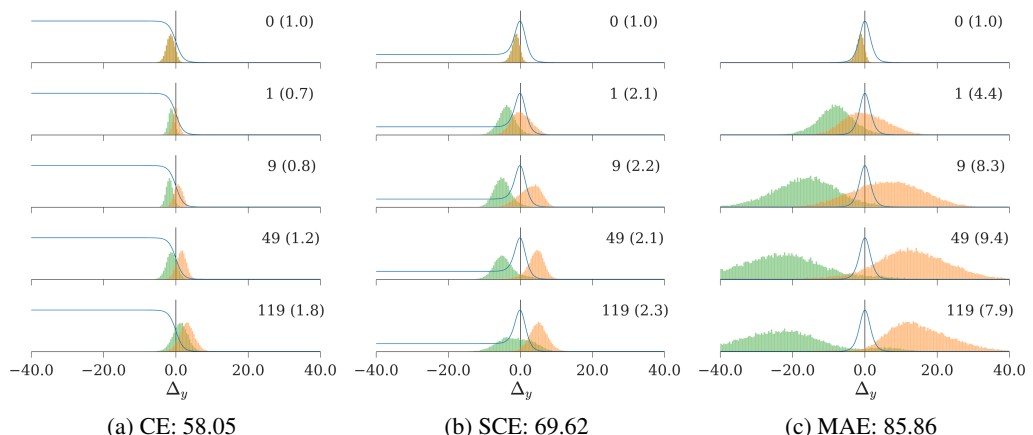

(a) CE: 58.05      (b) SCE: 69.62      (c) MAE: 85.86

Figure 4: How $\Delta_y$ distribution of noisy (green, left) and clean (orange, right) samples evolve during training on CIFAR10 with 40% symmetric label noise. We include $w(\Delta_y)$ curves for reference, and omit vertical axes denoting probability density for brevity. Vertical axes are scaled to the peak of histograms for better readability, with epoch number (axis scaling factor) denoted on the right of each subplot. We also include the final accuracy of the corresponding run for each loss function as reference. See Appendix F for results of more loss functions with human label noise.

### 4.3 Noise Robustness from a Sample-Weighting Curriculum Perspective

Intuitively, loss functions exhibiting noise robustness should weight clean samples more than noisy ones. We provide an explanation based on how $w(\Delta_y)$ interacts with two training dynamics.

**Robust losses assign larger weights to clean samples.** The average weight assigned to noisy samples during training, adjusted by learning rate $\alpha_t$, is $\bar{w}_{\text{noise}} = \sum_{i,t} \mathbb{I}(\tilde{y}_{i,t} = y_{i,t}) \alpha_t w_{i,t} / (\sum_{i,t} \mathbb{I}(\tilde{y}_{i,t} \neq y_{i,t}) \alpha_t)$, where $w_{i,t}$ denotes the weight of $i$-th sample of the batch at step $t$. $\bar{w}_{\text{clean}}$ for clean samples can be defined similarly. The ratio $\text{snr} = \bar{w}_{\text{clean}} / \bar{w}_{\text{noise}}$ characterizes their relative contribution during training. We report snr and the drop in test accuracy under different label noise on CIFAR10 in Table 4. Loss functions with less performance drop have higher snr in general.

To explain what leads to a high snr, we first examine how $\Delta_y$ distributions of noisy and clean samples evolve during training on CIFAR10 with symmetric label noise in Fig. 4. See Appendix F for results of more loss functions with human label noise. When trained using loss functions with increased robustness (Fig. 4b and 4c), the noisy and clean distributions of $\Delta_y$ gets better separated and more spread. In addition, $\Delta_y$'s of some noisy samples gets decreased, suggesting that noisy samples can be *unlearnt*. In contrast, with CE (Fig. 4a), the noisy and clean distributions of $\Delta_y$ are less separated and more compact.

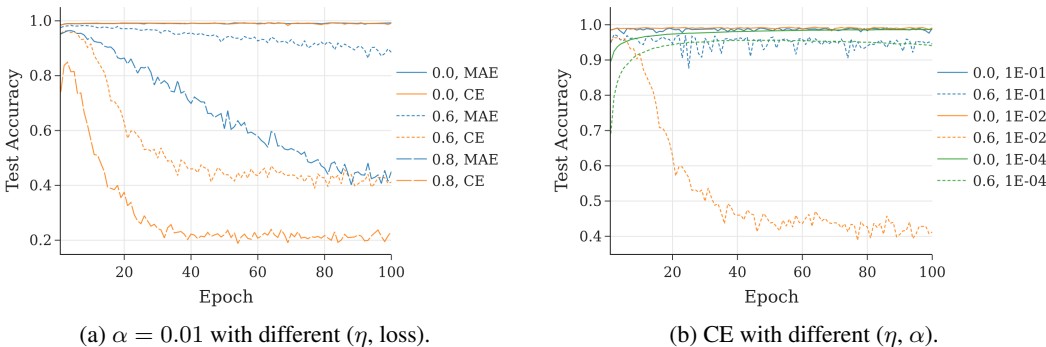

(a) $\alpha = 0.01$ with different $(\eta, \text{loss})$.

(b) CE with different $(\eta, \alpha)$.

Figure 5: Learning curves with fixed learning rate and extended training epochs on MNIST, where $\alpha$ is the learning rate and $\eta$ the symmetric label noise rate. Vertical axes are scaled for readability.

We now give a possible explanation for Fig. 4 with the following two training dynamics: **(D1) clean samples are learnt faster than noisy samples**; **(D2) noisy samples can be unlearnt when trained on clean samples.** D1 is identified in [7, 37], which later manifests itself in curriculum-based robust training [38, 39]. It can result from the dominance of clean samples ($\tilde{r} < 1$) in the expected gradient. In addition, gradients of clean samples are more correlated than those of noisy samples [40]. Thus performance on clean samples can be improved when training on one another, leading to D1. D2 only become apparent when examining Fig. 4b and 4c, which can result from generalization with clean samples. Suppose in MNIST, a sample of 0 is erroneously labeled as 9. Then a model well-trained with clean samples of class 9 and 0 can result in a low $\Delta_y$ for this noisy sample. D1 and D2 can act in synergy to separate the clean and noisy distributions of $\Delta_y$, as shown in Fig. 4.

We hypothesis that robust losses enhance the synergy of D1 and D2. In Table 1, $w(\Delta_y)$ of loss functions can be decomposed into $f(\Delta_y) \cdot g(\Delta_y)$, where $f(\Delta_y)$ is a monotonically increasing function and $g(\Delta_y)$ a decreasing one. For example, $f_{\text{CE}}(\Delta_y)$ degenerates to constant 1 and $g_{\text{CE}}(\Delta_y) = 1 - p_y$, while $f_{\text{MAE}}(\Delta_y) = p_y$ and $g_{\text{MAE}}(\Delta_y) = 1 - p_y$. Notably, $g(\Delta_y)$ shared by all loss functions converges to 0 as $\Delta_y$ increases, preventing $\Delta_y$ from growing infinitely large. In addition, **a non-degenerated $f(\Delta_y)$ can enhance the synergy between D1 and D2**. Since the initial $\Delta_y$ distribution generally lies on the monotonically increasing part of $w(\Delta_y)$ determined by $f(\Delta_y)$, faster learning of samples results in their larger weights during training. Thus robust losses **magnify the difference in learning speed** between clean and noisy samples, which can also account for the substantially spread $\Delta_y$ distributions in Fig. 4b and 4c. As $w(\Delta_y)$ can assign negligible sample weights with low $\Delta_y$ due to the monotonically increasing $f(\Delta_y)$, **unlearnt noisy samples are neglected** with diminishing weights, which can account for the decrease of $\Delta_y$'s for noisy samples in Fig. 4b and 4c. In contrast, as $w_{\text{CE}}(\Delta_y)$ assign high sample weights for small $\Delta_y$'s, it compensates the synergy of D1 and D2, thus results in compact $\Delta_y$ distribution, larger $\Delta_y$'s for noisy samples, and less separated $\Delta_y$ distributions in Fig. 4a.

With sufficient training, clean samples will eventually have high $\Delta_y$'s with diminishing sample weights thanks to $g(\Delta_y)$. Noisy samples will then dominate the expected gradient and can lead to overfitting, leading to two unexpected results when viewed from robustness conditions [9, 11]:

**Robust losses are vulnerable to label noise with extended training.** In Fig. 5a we show the learning curve of CE and MAE using *constant* learning rate under different symmetric noises on MNIST. Although enjoying theoretically guaranteed noise robustness [9, 11], similar to CE, MAE eventually overfits to noisy samples, becoming vulnerable to label noise as weights of clean samples diminish.

**Loss functions can become robust by adjusting the learning rate schedule.** Interestingly, in Fig. 4a, despite the compensation of $w_{\text{CE}}(\Delta_y)$, the synergy between D1 and D2 still results in partially-separated $\Delta_y$ distributions of noisy and clean samples. We can thus improve the noise robustness of CE by preventing the weights of clean samples from diminishing due to $g(\Delta_y)$, which can be achieve by slowing down the convergence or early stopping [41]. In Fig. 5b we show the learning curve of CE using fixed learning rates under symmetric noise on MNIST. By simply increasing or decreasing the learning rate, which strengthens the implicit regularization of SGD [42] or directly slows down the convergence, the noise robustness of CE can be substantially improved.

## 5 Related Work

Our work is closely related to robust loss functions [8–13] for robust training with noisy labels [1]. Theoretical results [9, 11] derive sufficient conditions for robustness against label noise without considering the training dynamics. We complement these results by considering the interaction between robust losses and various training dynamics. The underfitting of robust losses has been heuristically mitigated with loss combination [10, 12, 13]. We further elucidate the cause of underfitting from a curriculum perspective, based on which we provide an effective solution.

Curriculum-based approaches combat label noise with either sample selection [21, 22] or sample-weighting [18–20]. In particular, sample weights are designed [16–18] or predicted by a model trained on a separated dataset [19, 20]. In contrast, the sample-weighting curriculums considered in this work are implicitly induced by robust loss functions. Most related to our work, Wang et al. [16] identifies gradient norms as weights for sample gradients of each robust loss. In contrast, as discussed in §3.1, we explicitly extract the implicit loss, which helps draw the connection to standard curriculum learning [24] and facilitates analysis of training dynamics.

Our work is also related to the ongoing debate [24, 43] on strategies for selecting or weighting samples in curriculum learning: whether easier first [23, 26] or harder first [27, 44] is better. The implicit curriculums of robust losses in this work differ in two important ways. First, the implicit loss identified in §3.1 more directly measures sample difficulty than loss value [26, 27] and gradient magnitude [28]. Second, the implicit sample-weighting curriculums can be viewed as a combination of both weighting strategies by emphasizing moderately difficult samples, as discussed in §4.3.

## 6 Conclusion

We identify the implicit sample-weighting curriculums of selected loss functions. By decoupling the implicit loss as a direct sample performance metric and sample weights specifying the implicit sample preference, we can analyze how robust loss functions and curriculums interact with different training dynamics. Such a perspective complements existing research on theoretical bounds for the risk minimizer, and connects robust loss functions to the seemingly distinct approaches based on curriculum learning. Following the curriculum perspective, we elucidate the reasons behind underfitting and robustness against label noise for existing robust loss functions, and design a simple approach to address the underfitting issue.

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

# A   Extended Review of Loss Functions

As a general reference, we provide an extended review of loss functions for classification that is relevant to the standard form Eq. (7), complementing review in §2.2. Loss functions and their sample-weighting functions are summarized in Table 5. We plot how hyperparameters affect their sample-weighting functions in Fig. 6.

## A.1   Loss Functions without Robustness Guarantees

**Cross Entropy (CE)**
$$L_{\mathrm{CE}}(\boldsymbol{s}, y) = -\log p_y$$
is the standard loss function for classification.

**Focal Loss (FL)** [45]
$$L_{\mathrm{FL}}(\boldsymbol{s}, y) = -(1 - p_y)^q \log p_y$$
aims to address the label imbalance in object detection. Note that both CE and FL are neither symmetric [10] nor asymmetric [11].

## A.2 Symmetric Losses

**Mean Absolute Error (MAE)** [9]

$$L_{\mathrm{MAE}}(\boldsymbol{s}, y) = \frac{1}{k} \sum_{i=1}^{k} |\mathbb{I}(i = y) - p_i| = 2 - 2p_y \propto 1 - p_y$$

is a classic symmetric loss, where $\mathbb{I}(i = y)$ is the indicator function.

**Reverse Cross Entropy (RCE)** [13]

$$L_{\mathrm{RCE}}(\boldsymbol{s}, y) = \sum_{i=1}^{k} p_i \log \mathbf{1}(i = y) = \sum_{i \neq y} p_i A = (1 - p_y) A \propto 1 - p_y = L_{\mathrm{MAE}}(\boldsymbol{s}, y)$$

is equivalent to MAE in implementation, where $\log 0$ is truncated to a negative constant $A$ to avoid numerical overflow.

Ma et al. [10] argued that any generic loss functions with $L(\boldsymbol{s}, i) > 0, \forall i \in \{1, \ldots, k\}$ can become symmetric by simply normalizing them. As an example,

**Normalized Cross Entropy (NCE)**

$$L_{\mathrm{NCE}}(\boldsymbol{s}, y) = \frac{L_{\mathrm{CE}}(\boldsymbol{s}, y)}{\sum_{i=1}^{k} L_{\mathrm{CE}}(\boldsymbol{s}, i)} = \frac{-\log p_y}{\sum_{i=1}^{k} -\log p_i}$$

is a symmetric loss [10]. However, NCE does not follow the standard form of Eq. (7). It involves an additional regularizer as discussed in §3.2 and Appendix C, thus being more relevant to discussions in Appendix A.4.

## A.3 Asymmetric Losses

Zhou et al. [11] derived the asymmetric condition for noise robustness, and propose an array of asymmetric losses:

**Asymmetric Generalized Cross Entropy (AGCE)**

$$L_{\mathrm{AGCE}}(\boldsymbol{s}, y) = \frac{(a + 1) - (a + p_y)^q}{q}$$

where $a > 0$ and $q > 0$. It is asymmetric when $\mathbb{I}(q \leq 1)(\frac{a+1}{a})^{1-q} + \mathbb{I}(q > 1) \leq 1/\tilde{r}$.

**Asymmetric Unhinged Loss (AUL)**

$$L_{\mathrm{AUL}}(\boldsymbol{s}, y) = \frac{(a - p_y)^q - (a - 1)^q}{q}$$

where $a > 1$ and $q > 0$. It is asymmetric when $\mathbb{I}(q \leq 1)(\frac{a}{a-1})^{q-1} + \mathbb{I}(q \leq 1) \leq 1/\tilde{r}$.

**Asymmetric Exponential Loss (AEL)**

$$L_{\mathrm{AEL}}(\boldsymbol{s}, y) = e^{-p_y/q}$$

where $q > 0$. It is assymetric when $e^{1/q} \leq 1/\tilde{r}$.

### A.3.1 Combined Losses

Loss functions can be combined to enjoy better learning.

**Generalized Cross Entropy (GCE)** [12]

$$L_{\mathrm{GCE}}(\boldsymbol{s}, y) = \frac{1 - p_y^q}{q}$$

can be viewed as a smooth interpolation between CE and MAE, where $0 < q \leq 1$. CE or MAE can be recovered by setting $q \to 0$ or $q = 1$.

**Symmetric Cross Entropy (SCE)** [13]

$$L_{\mathrm{SCE}}(\boldsymbol{s}, y) = a \cdot L_{\mathrm{CE}}(\boldsymbol{s}, y) + b \cdot L_{\mathrm{RCE}}(\boldsymbol{s}, y) \propto (1 - q) \cdot (-\log p_i) + q \cdot (1 - p_i)$$

| Name | Function | Sample Weight $w$ | Constraints |
|---|---|---|---|
| CE | $-\log p_y$ | $1 - p_y$ | |
| FL | $-(1-p_y)^q \log p_y$ | $(1-p_y)^q(1-p_y-qp_y\log p_y)$ | $q > 0$ |
| MAE/RCE | $1 - p_y$ | $p_y(1-p_y)$ | |
| AUL | $\frac{(a+1)-(a+p_y)^q}{q}$ | $p_y(1-p_y)(a-p_y)^{q-1}$ | $a > 1, q > 0$ |
| AGCE | $\frac{(a-p_y)^q-(a-1)^q}{q}$ | $p_y(a+p_y)^{q-1}(1-p_y)$ | $a > 0, q > 0$ |
| AEL | $e^{-p_y/q}$ | $\frac{1}{q}p_y(1-p_y)e^{-p_y/q}$ | $q > 0$ |
| GCE | $(1-p_y^q)/q$ | $p_y^q(1-p_y)$ | $0 < q \leq 1$ |
| SCE | $-(1-q)\log p_y + q(1-p_y)$ | $(1-q+q\cdot p_y)(1-p_y)$ | $0 < q < 1$ |
| TCE | $\sum_{i=1}^{q}(1-p_y)^i/i$ | $p_y\sum_{i=1}^{q}(1-p_y)^i$ | $q \geq 1$ |

Table 5: Expressions, constraints of hyperparameters and sample-weighting functions of loss functions in Appendix A that follows the standard form Eq. (7).

is a weighted average of CE and RCE (MAE), where $a > 0$, $b > 0$, and $0 < q < 1$.

**Taylor Cross Entropy (TCE)** [15]

$$L_{\mathrm{TCE}}(\boldsymbol{s}, y) = \sum_{i=1}^{q} \frac{(1-p_y)^i}{i}$$

is originally derived from Taylor series of the $\log$ function. TCE reduces to MAE when $q = 1$. Interestingly, the summand of TCE $(1-p_y)^i/i$ with $i > 2$ is proportional to AUL with $a = 1$ and $q = i$. Thus TCE can be viewed as a combination of symmetric and asymmetric losses.

Ma et al. [10] propose to additively combine active and passive loss functions. We review NCE+MAE as an example:

$$L_{\mathrm{NCE+MAE}}(\boldsymbol{s}, y) = a \cdot L_{\mathrm{NCE}}(\boldsymbol{s}, y) + b \cdot L_{\mathrm{MAE}}(\boldsymbol{s}, y) \propto (1-q) \cdot \frac{-\log p_y}{\sum_{i=1}^{k} -\log p_i} + q \cdot (1-p_y)$$

where $a > 0$, $b > 0$, and $0 < q < 1$.

### A.4 Loss Functions with Additional Regularizers

We additionally review loss functions that implicitly involve a regularizer and a primary loss function that fits the standard form Eq. (7). See Table 6 for a summary. We leave investigation on how these additional regularizers affect noise robustness for future work.

**Mean Square Error (MSE)** [9]

$$L_{\mathrm{MSE}}(\boldsymbol{s}, y) = \sum_{i=1}^{k}(\mathbb{I}(i=y)-p_i)^2 = 1 - 2p_y + \sum_{i=1}^{k} p_i^2$$

$$\propto 1 - p_y + \frac{1}{2} \cdot \sum_{i=1}^{k} p_i^2 = L_{\mathrm{MAE}}(\boldsymbol{s}, y) + \alpha \cdot R_{\mathrm{MSE}}(\boldsymbol{s})$$

is argued [9] to be more robust than CE, where $\alpha = \frac{1}{2}$ and the regularizer

$$R_{\mathrm{MSE}}(\boldsymbol{s}) = \sum_{i=1}^{k} p_i^2 \tag{14}$$

reduces the entropy of the softmax output. We can generalize $\alpha$ to a hyperparamter, making MSE a combination of MAE and an entropy regularizer $R_{\mathrm{MSE}}$.

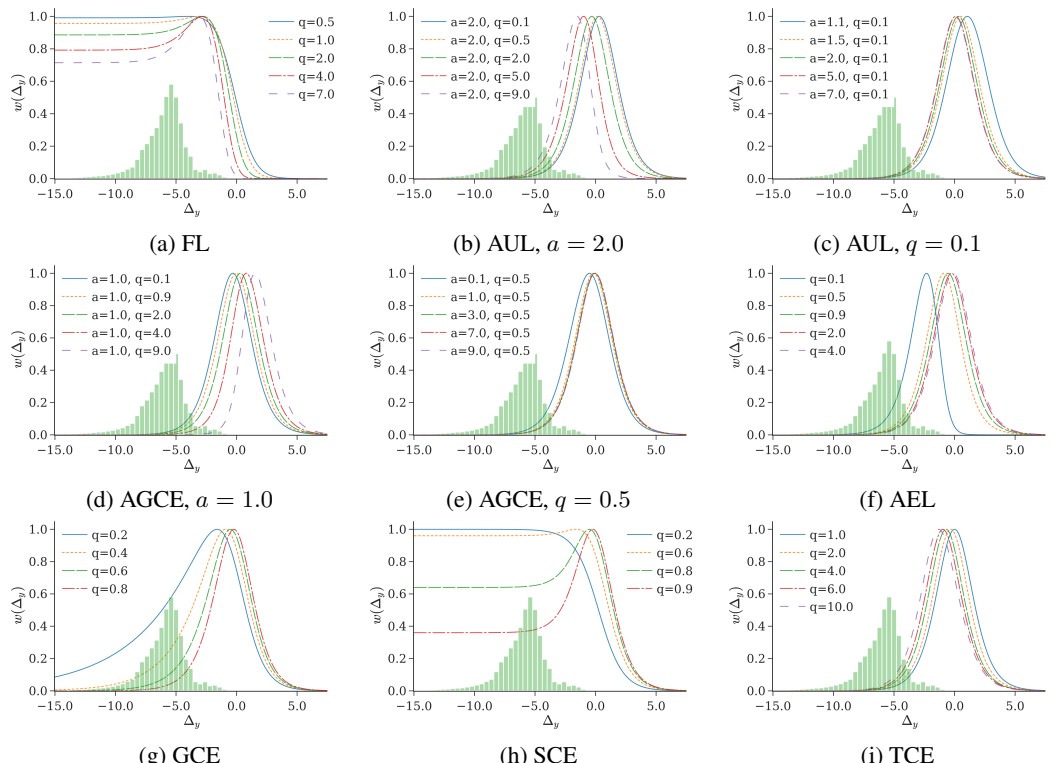

Figure 6: How hyperparameters affect the sample-weighting functions of loss functions in Table 5. The initial $\Delta_y$ distribution of CIFAR100 are included as reference.

Given a generic loss function $L(\boldsymbol{s}, y)$, **Peer Loss (PL)** [14]

$$L_{\mathrm{PL}}(\boldsymbol{s}, y) = L(\boldsymbol{s}, y) - L(\boldsymbol{s}_{n_1}, y_{n_2})$$

can make it robust against label noise, where $\boldsymbol{s}_{n_1}$ and $y_{n_2}$ denote scores (of input $\boldsymbol{x}_{n_1}$) and labels randomly sampled from the noisy data. PL is inspired by the peer prediction mechanism to truthfully elicit information when there is no ground truth verification. Its noise robustness is theoretically established for binary classification and extended to multi-class setting [14]. Cheng et al. [46] later show that PL in its expectation is equivalent to the original loss plus a **Confidence Regularizer (CR)**:

$$R_{\mathrm{CR}}(\boldsymbol{s}) = -\mathbb{E}_{\tilde{y}}[L(\boldsymbol{s}, \tilde{y})]$$

Substituting $L$ with the standard $L_{\mathrm{CE}}$, $R_{\mathrm{CR}}(\boldsymbol{s})$ becomes

$$R_{\mathrm{CR}}(\boldsymbol{s}) = -\mathbb{E}_{\tilde{y}}[-\log p_{\tilde{y}}] = \sum_{i=1}^{k} P(\tilde{y} = i) \log p_i \tag{15}$$

Minimizing $R_{\mathrm{CR}}(\boldsymbol{s})$ thus makes the softmax output distribution $p_i$'s deviate from the prior label distribution of the noisy dataset $P(\tilde{y} = i)$'s, reducing the entropy of the softmax output.

Label smoothing [47] has been shown to mitigate overfitting with label noise [48]. With the standard cross entropy, **Generalized Label Smoothing (GLS)** [49]

$$L_{\mathrm{GLS+CE}}(\boldsymbol{s}, y) = \sum_{i=1}^{k} -[\mathbb{I}(i = y)(1 - \alpha) + \frac{\alpha}{k}] \log p_i$$

$$= -(1 - \alpha) \log p_y - \alpha \cdot \frac{1}{k} \sum_{i=1}^{k} \log p_i$$

$$\propto -\log p_y - \frac{\alpha}{1 - \alpha} \cdot \frac{1}{k} \sum_{i=1}^{k} \log p_i = L_{\mathrm{CE}}(\boldsymbol{s}, y) + \alpha' \cdot R_{\mathrm{GLS}}(\boldsymbol{s})$$

| Name | Original | Primary Loss | Regularizer |
|------|----------|--------------|-------------|
| MSE | $1 - 2p_y + \sum_{i=1}^{k} p_i^2$ | $1 - p_y$ | $\sum_{i=1}^{k} p_i^2$ |
| PL | $-\log p_y + \log p_{y_{n_2}\mid \boldsymbol{x}_{n_1}}$ | $-\log p_y$ | $\sum_{i=1}^{k} P(\tilde{y}=i) \log p_i$ |
| GLS | $-\sum_{i=1}^{k}[\mathbb{I}(i=y)(1-\alpha) + \frac{\alpha}{k}]\log p_i$ | $-\log p_y$ | $\pm \sum_{i=1}^{k} \frac{1}{k}\log p_i$ |
| NCE | $\dfrac{-\log p_y}{\sum_{i=1}^{k} -\log p_i}$ | $\text{stop\_grad}\left(\frac{1}{\sum_{i=1}^{k}\log p_i}\right)\log p_i$ | $\sum_{i=1}^{k}\frac{1}{k}\log p_i$ |

Table 6: Original expressions, primary losses following the standard form Eq. (7) and regularizers for loss functions reviewed in Appendix A.4. We view PL in its expectation to derive its regularizer. $p_{y_{n_2}\mid \boldsymbol{x}_{n_1}}$ is the softmax output with a random input $\boldsymbol{x}_{n_1}$ and a random label $y_{n_2}$ sampled from the noisy data.

| Loss | CIFAR10 | CIFAR100 |
|------|---------|----------|
| SCE | $q = 0.7$ | $q = 0.95$ |
| GCE | $q = 0.3$ | $q = 0.9$ |
| NCE+MAE | $q = 0.3$ | $q = 0.9$ |
| AUL | $a = 1.1, q = 5$ | $a = 7.0, q = 0.5$ |
| AGCE | $a = 0.1, q = 0.1$ | $a = 3.0, q = 1.2$ |
| AUL$^\dagger$ | $a = 3.0, q = 0.7$ | / |
| AGCE$^\dagger$ | $a = 1.6, q = 2.0$ | / |
| FL | / | $q = 2$ |
| AEL | / | $q = 1.5$ |
| TCE | / | $q = 6$ |

Table 7: Hyperparameters of each loss function on different datasets. AUL$^\dagger$ and AGCE$^\dagger$ are with inferior hyperparameters.

where $\alpha' = \alpha/(1-\alpha)$, has regularizer $R_{\text{GLS}}$

$$R_{\text{GLS}}(\boldsymbol{s}) = -\sum_{i=1}^{k} \frac{1}{k}\log p_i \tag{16}$$

With $\alpha' > 0$, $R_{\text{GLS}}$ corresponds to the original label smoothing [47], which increases the entropy of softmax outputs. In contrast, $\alpha' < 0$ corresponding to negative label smoothing [49], which decreases the output entropy similar to $R_{\text{CR}}$.

Finally, with equivalent derivatives, **NCE** discussed in §3.2 and Appendix C can be decomposed into

$$L_{\text{NCE}}(\boldsymbol{s}, y) = \frac{1}{\sum_{i=1}^{k} -\log p_i} \left\{ -\log p_y + \frac{k\log p_y}{\sum_{i=1}^{k}\log p_i}\cdot\left[\frac{1}{k}\sum_{i=1}^{k}\log p_i\right]\right\}$$
$$= \text{stop\_grad}(\gamma_{\text{NCE}})\cdot[L_{\text{CE}}(\boldsymbol{s}, y) + \text{stop\_grad}(\epsilon_{\text{NCE}})\cdot R_{\text{NCE}}(\boldsymbol{s})]$$

where

$$R_{\text{NCE}}(\boldsymbol{s}) = \sum_{i=1}^{k}\frac{1}{k}\log p_i \tag{17}$$

is the same regularizer as $R_{\text{GLS}}$ with a negative weight $-\epsilon_{\text{NCE}}$.

# B  Detailed Experimental Settings

Our settings follow [10, 11], with differences explicitly stated in the main text. All models on CIFAR10/100 and MNIST are trained on NVIDIA 2080ti gpus with FP32. For models on the large scale dataset WebVision [36], we use FP16 to accelerate training.

**Synthetic noise generation**  The noisy labels are generated following [10, 11, 50]. For symmetric label noise, the training labels are randomly flipped to a different class with with probabilities

$\eta \in \{0.2, 0.4, 0.6, 0.8\}$. Asymmetric label noise are generated by a class-dependent flipping pattern. On CIFAR-100, the 100 classes are grouped into 20 super-classes each having 5 sub-classes. Each class are flipped within the same super-class into the next in a circular fashion. The flip probabilities are $\eta \in \{0.1, 0.2, 0.3, 0.4\}$.

**Models and Training**  We use a 4-layer CNN for MNIST, an 8-layer CNN for CIFAR10, a ResNet-34 [32] for CIFAR100, and a ResNet-50 [32] for WebVision, all with batch normalization [34]. Data augmentation including random width/height shift and horizontal flip are applied to CIFAR10/100. On WebVision, we additionally include random cropping and color jittering. Without further specifications, all models are trained using SGD with momentum 0.9 and batch size 128 for 50, 120, 200 and 250 epochs on MNIST, CIFAR10, CIFAR100 and WebVision, respectively. Learning rates with cosine annealing are 0.01 on MNIST and CIFAR10, 0.1 on CIFAR100, and 0.2 on WebVision. Weight decays are $10^{-3}$ on MNIST, $10^{-4}$ on CIFAR10, $10^{-5}$ on CIFAR100 and $3 \times 10^{-5}$ on WebVision. Notably, all loss functions are normalized to have unit maximum in sample weights, which is different from [10]. See Table 7 for hyperparameters of loss functions on different datasets.

## C  Deriving the Upperbound of Sample Weights of NCE

We provide detailed derivations for results in §3.2.

**Constant Norm of** $\|\nabla_{\boldsymbol{s}}\Delta_y\|_1$: Since

$$\frac{\partial \Delta_y}{\partial s_i} = \begin{cases} 1, & i = y \\ -\frac{e^{s_i}}{\sum_{k \neq y} e^{s_k}} = \frac{p_i}{1 - p_y}, & i \neq y \end{cases}$$

then

$$\|\nabla_{\boldsymbol{s}}\Delta_y\|_1 = \sum_i |\frac{\partial \Delta_y}{\partial s_i}| = 1 + \sum_{i \neq y} \frac{e^{s_i}}{\sum_{k \neq y} e^{s_k}} = 1 + 1 = 2$$

**Approximating upperbound of** $w_{\mathrm{NCE}}$ **in Eq. (10)**:

$$\begin{aligned}
w_{\mathrm{NCE}} &= \frac{\|\nabla_{\boldsymbol{s}} L_{\mathrm{NCE}}(\boldsymbol{s}, y)\|_1}{\|\nabla_{\boldsymbol{s}} \Delta_y\|_1} = \frac{1}{2}\|\nabla_{\boldsymbol{s}} L_{\mathrm{NCE}}(\boldsymbol{s}, y)\|_1 \\
&\leq \frac{1}{2}\gamma_{\mathrm{NCE}} \cdot (\|\nabla_{\boldsymbol{s}} L_{\mathrm{CE}}(\boldsymbol{s}, y)\|_1 + \epsilon_{\mathrm{NCE}} \cdot \|\nabla_{\boldsymbol{s}} R_{\mathrm{CE}}(\boldsymbol{s})\|_1) \\
&\leq \frac{1}{2}\gamma_{\mathrm{NCE}} \cdot \left(\|\nabla_{\boldsymbol{s}} L_{\mathrm{CE}}(\boldsymbol{s}, y)\|_1 + \epsilon_{\mathrm{NCE}} \cdot \frac{1}{k} \sum_j \|\nabla_{\boldsymbol{s}} L_{\mathrm{CE}}(\boldsymbol{s}, j)\|_1\right) \\
&= \gamma_{\mathrm{NCE}} \left(w_{\mathrm{CE}} + \frac{k-1}{k}\epsilon_{\mathrm{NCE}}\right)
\end{aligned}$$

The derivation is based on the inequality $|x \pm y| \leq |x| + |y|$, following the intuition [16, 17] that $\|\nabla_{\boldsymbol{s}} L_{\mathrm{NCE}}(\boldsymbol{s}, y)\|_1$ can be regarded as sample weights.

## D  Underfitting of Robust Losses: Additional Results

In Table 8 we report similar results as Table 2 in §4.1 with smaller learning rates. Although settings that severe underfits slightly improve, the performance gap compared to CE is still substantial. Such results further confirms that underfitting results from robust losses themselves.

## E  Fixing Underfitting: Derivations and Additional Results

We include detailed derivations and additional results for §4.2.

**Simulated** $\Delta_y$**'s well approximate real settings.** We compare the simulated $\Delta_y$ distributions to that of real datasets at initialization in Fig. 7. Although less accurate with the variance, the simulated expectations mostly follow real settings, which supports the analysis in §4.2.

| Underfitting | Loss | CIFAR100 Acc. | $\bar{\alpha}_t^*$ | CIFAR10 Acc. | $\bar{\alpha}_t^*$ |
|---|---|---|---|---|---|
| No | CE | $68.76 \pm 0.21$ | 0.962 | $90.24 \pm 0.14$ | 0.624 |
| No | GCE | $69.00 \pm 0.24$ | 0.956 | $90.83 \pm 0.20$ | 0.644 |
| | SCE | $68.89 \pm 0.05$ | 1.165 | $91.07 \pm 0.09$ | 0.726 |
| | NCE+MAE | $68.21 \pm 0.51$ | 0.520 | $90.14 \pm 0.09$ | 0.344 |
| Moderate | NCE | $57.95 \pm 0.26$ | 0.330 | $85.96 \pm 0.21$ | 0.206 |
| | AUL | $47.98 \pm 3.48$ | 0.485 | $88.94 \pm 0.29$ | 0.604 |
| | AGCE | $43.51 \pm 2.58$ | 0.406 | $90.71 \pm 0.19$ | 0.549 |
| Severe | MAE | $9.11 \pm 0.83$ | 0.025 | $90.65 \pm 0.10$ | 0.355 |
| | AUL$^\dagger$ | $10.04 \pm 2.33$ | 0.023 | $90.77 \pm 0.04$ | 0.337 |
| | AGCE$^\dagger$ | $5.34 \pm 0.67$ | 0.008 | $81.59 \pm 8.55$ | 0.243 |

Table 8: Similar results as Table 2 except with learning rate $\alpha = 0.01$. See Table 7 for detailed hyperparameters. AUL$^\dagger$ and AGCE$^\dagger$ with inferior hyperparamters are included as reference. Robust losses can underfit regardless of hyperparameters of training.

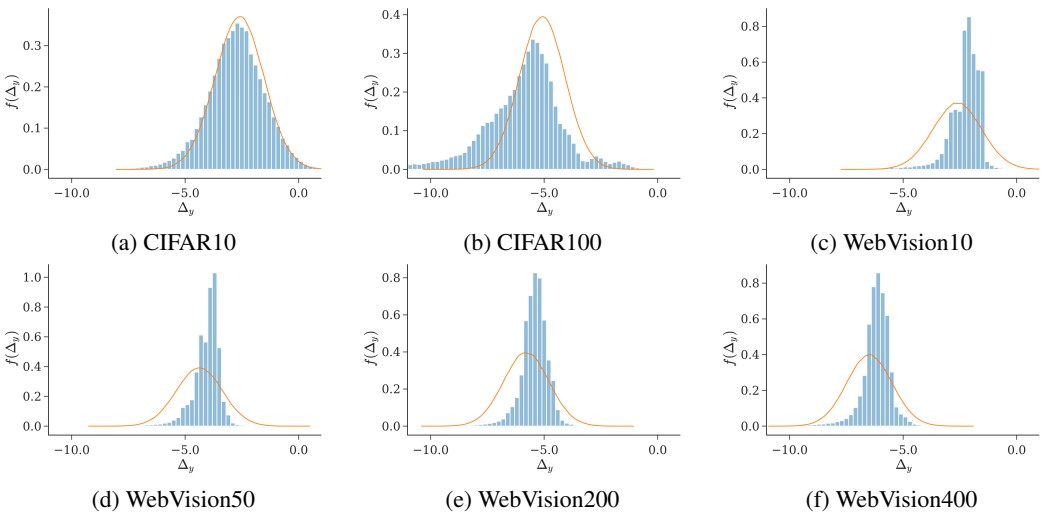

Figure 7: Comparing simulated and real $\Delta_y$ distributions at initialization. We simulate with class scores following standard normal distribution, i.e., $s_i \sim \mathcal{N}(0, 1)$. Histograms are real distributions while the curves are from simulations, with the vertical axis denoting probability density $f(\Delta_y)$.

**Deriving $\mathbb{E}(\Delta_y)$ in Eq. (11)** :

$$\mathbb{E}(\Delta_y) = \mathbb{E}[s_y - \log \sum_{j \neq y} e^{s_j}] = \mu - \mathbb{E}[\log \sum_{j \neq y} e^{s_j}]$$

$$\approx_1 \mu - \log \mathbb{E}[\sum_{j \neq y} e^{s_j}] + \frac{\mathbb{V}[\sum_{j \neq y} e^{s_j}]}{2\mathbb{E}[\sum_{j \neq y} e^{s_j}]^2}$$

$$=_2 \mu - \log\{(k-1)\mathbb{E}[e^{s_y}]\} + \frac{(k-1)\mathbb{V}[e^{s_y}]}{2\{(k-1)\mathbb{E}[e^{s_y}]\}^2}$$

$$=_3 \mu - \log[(k-1)e^{\mu+\sigma^2/2}] + \frac{(k-1)(e^{\sigma^2}-1)e^{2\mu+\sigma^2}}{2[(k-1)e^{\mu+\sigma^2/2}]^2}$$

$$= -\log(k-1) - \sigma^2/2 + \frac{e^{\sigma^2}-1}{2(k-1)}$$

where $\approx_1$ follows $\mathbb{E}[\log X] \approx \log \mathbb{E}[X] - \frac{\mathbb{V}[X]}{2\mathbb{E}[X]^2}$ [51], $=_2$ utilize properties of sum of log-normal variables [52], and $=_3$ substitutes the expression of $\mathbb{E}[e^{s_y}]$ and $\mathbb{V}[e^{s_y}]$ for log-normal distributions.

| Loss | Clean $\eta = 0$ | Symmetric Noise (Noise Rate $\eta$) | | | |
|---|---|---|---|---|---|
| | | $\eta = 0.2$ | $\eta = 0.4$ | $\eta = 0.6$ | $\eta = 0.8$ |
| CE [11] | $71.33 \pm 0.43$ | $56.51 \pm 0.39$ | $39.92 \pm 0.10$ | $21.39 \pm 1.17$ | $7.59 \pm 0.20$ |
| GCE [11] | $63.09 \pm 1.39$ | $61.57 \pm 1.06$ | $56.11 \pm 1.35$ | $45.28 \pm 0.61$ | $17.42 \pm 0.06$ |
| NCE [11] | $29.96 \pm 0.73$ | $25.27 \pm 0.32$ | $19.54 \pm 0.52$ | $13.51 \pm 0.65$ | $8.55 \pm 0.37$ |
| NCE+AUL [11] | $68.96 \pm 0.16$ | $65.36 \pm 0.20$ | $59.25 \pm 0.23$ | $46.34 \pm 0.21$ | $23.03 \pm 0.64$ |
| AGCE | $49.27 \pm 1.03$ | $49.17 \pm 2.15$ | $47.76 \pm 1.75$ | $38.17 \pm 1.43$ | $16.03 \pm 0.59$ |
| AGCE shift | $67.50 \pm 1.48$ | $61.95 \pm 1.48$ | $53.33 \pm 1.08$ | $33.26 \pm 0.37$ | $10.47 \pm 0.57$ |
| AGCE rescale | $67.20 \pm 0.79$ | $64.28 \pm 1.27$ | $56.32 \pm 0.59$ | $38.52 \pm 1.67$ | $12.75 \pm 1.10$ |
| MAE | $3.69 \pm 0.59$ | $2.92 \pm 0.46$ | $1.29 \pm 0.50$ | $2.27 \pm 1.24$ | $1.00 \pm 0.00$ |
| MAE shift | $69.02 \pm 0.78$ | $59.75 \pm 0.84$ | $44.60 \pm 0.24$ | $24.27 \pm 0.26$ | $8.08 \pm 0.26$ |
| MAE rescale | $69.95 \pm 1.21$ | $66.42 \pm 0.71$ | $60.70 \pm 0.30$ | $45.17 \pm 2.37$ | $10.79 \pm 0.97$ |

Table 9: Shifting or rescaling $\Delta_y$ mitigates underfitting on CIFAR100 with symmetric label noise. We use $a = 2.6$ for MAE and AGCE and $a = 4.5$ for AGCE. Test accuracies are reported with 3 different runs. We also include results from [11] as context.

| Loss | Clean $\eta = 0$ | Asymmetric Noise (Noise Rate $\eta$) | | | |
|---|---|---|---|---|---|
| | | $\eta = 0.1$ | $\eta = 0.2$ | $\eta = 0.3$ | $\eta = 0.4$ |
| CE [11] | $71.33 \pm 0.43$ | $64.85 \pm 0.37$ | $58.11 \pm 0.32$ | $50.68 \pm 0.55$ | $40.17 \pm 1.31$ |
| GCE [11] | $63.09 \pm 1.39$ | $63.01 \pm 1.01$ | $59.35 \pm 1.10$ | $53.83 \pm 0.64$ | $40.91 \pm 0.57$ |
| NCE [11] | $29.96 \pm 0.73$ | $27.59 \pm 0.54$ | $25.75 \pm 0.50$ | $24.28 \pm 0.80$ | $20.64 \pm 0.40$ |
| NCE+AUL [11] | $68.96 \pm 0.16$ | $66.62 \pm 0.09$ | $63.86 \pm 0.18$ | $50.38 \pm 0.32$ | $38.59 \pm 0.48$ |
| AGCE | $49.27 \pm 1.03$ | $47.53 \pm 0.73$ | $46.77 \pm 2.37$ | $39.82 \pm 2.70$ | $33.40 \pm 1.57$ |
| AGCE-shift | $67.50 \pm 1.48$ | $64.07 \pm 0.90$ | $56.16 \pm 1.44$ | $46.73 \pm 1.39$ | $38.37 \pm 1.55$ |
| AGCE-rescale | $67.20 \pm 0.79$ | $65.69 \pm 0.24$ | $60.80 \pm 0.77$ | $48.72 \pm 1.39$ | $40.00 \pm 0.27$ |
| MAE | $3.69 \pm 0.59$ | $3.59 \pm 0.56$ | $3.19 \pm 0.98$ | $2.11 \pm 1.93$ | $2.53 \pm 1.34$ |
| MAE-shift | $69.02 \pm 0.78$ | $63.82 \pm 0.84$ | $56.38 \pm 0.45$ | $48.93 \pm 0.53$ | $40.57 \pm 0.47$ |
| MAE-rescale | $69.95 \pm 1.21$ | $68.01 \pm 1.08$ | $65.71 \pm 0.47$ | $57.40 \pm 0.35$ | $39.22 \pm 1.54$ |

Table 10: Shifting or rescaling $\Delta_y$ mitigates underfitting on CIFAR100 with asymmetric label noise. We use $a = 2.6$ for MAE and AGCE and $a = 4.5$ for AGCE. Test accuracies are reported with 3 different runs. We also include results from [11] as context.

**Additional results of shifted and rescaled fix to robust losses.** We report results with symmetric (Table 9) and asymmetric (Table 10) label noise with diverse noise rates $\eta$. For real world noisy datasets, we subsample WebVision following standard settings [10, 11] with different number of classes, and report results with MAE and ResNet50 in Table 11. See Appendix B for detailed experimental settings. Notably, WebVision50 corresponds to the mini setting adopted in previous work [10, 11]. Shift and rescale $\Delta_y$ mitigate underfitting of MAE and AGCE in general, resulting in performance similar to the state-of-the-arts.

# F Understanding Robustness: Additional Results

As a more extended exploration to Fig. 4 in §4.3, in Fig. 8 we plot how distribution of $\Delta_y$ evolve with more loss functions and more number of epochs on human label noise of CIFAR10-N [31]. They all follow similar trends as in Fig. 4.

|  | $\begin{array}{c} k = 10 \\ a = 2.2 \end{array}$ | $\begin{array}{c} k = 50 \\ a = 2.0 \end{array}$ | $\begin{array}{c} k = 200 \\ a = 1.8 \end{array}$ | $\begin{array}{c} k = 400 \\ a = 1.6 \end{array}$ |
|---|---|---|---|---|
| CE | 62.40 | 66.40 | 70.26 | / |
| MAE | 10.0 | 3.68 | 0.50 | / |
| MAE-shift | 58.40 | 60.76 | 59.31 | / |
| MAE-rescale | 48.40 | 66.72 | 71.92 | / |

Table 11: Shifting or rescaling $\Delta_y$ mitigates underfitting on real noisy dataset WebVision [36] with different number of classes. Due to the scale of the dataset, we only report test accuracy with a single run.

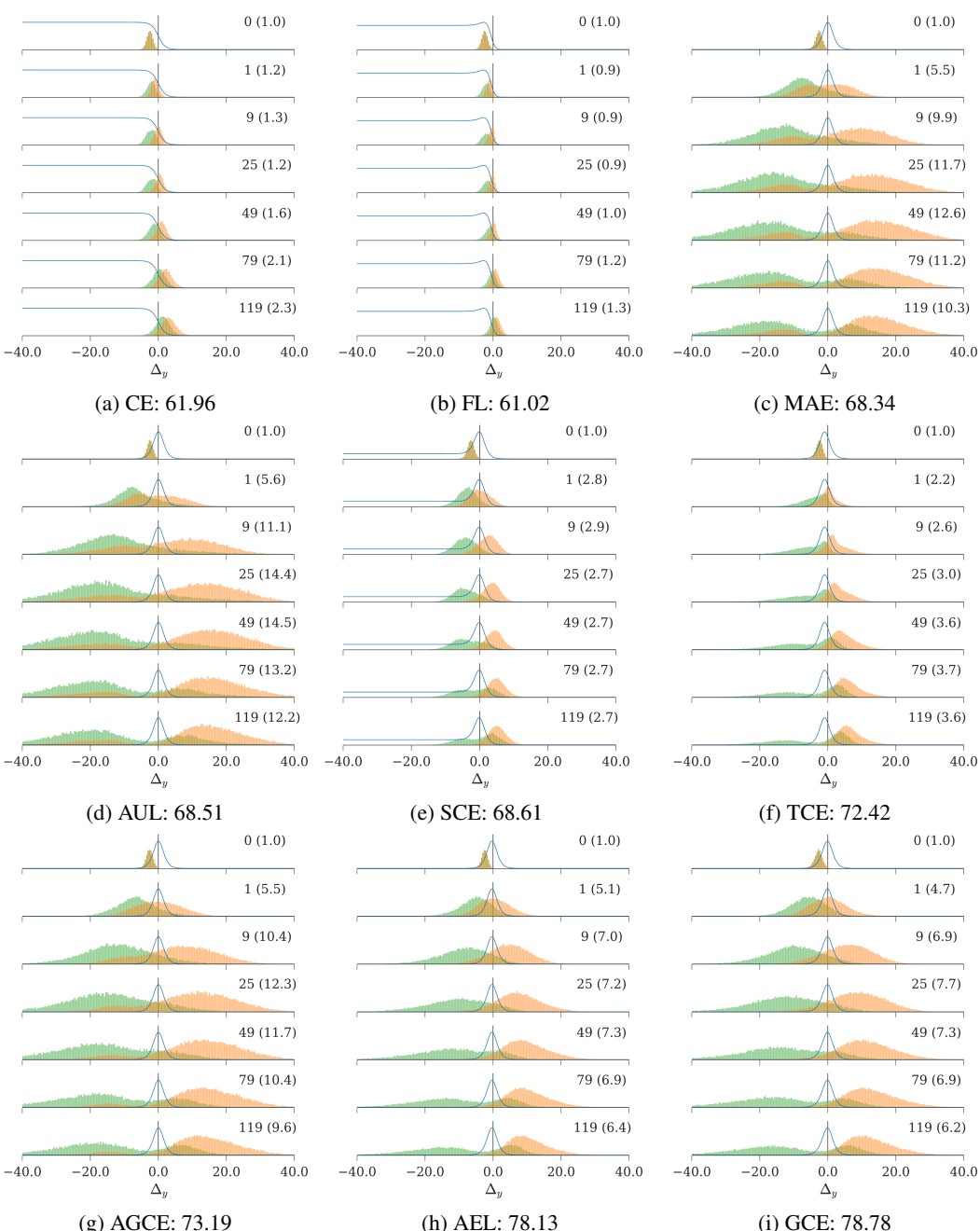

Figure 8: Additional results as Fig. 4 for more loss functions in Table 5 on CIFAR10-N [31] with "worst" noisy labels ($\eta = 0.4$). Note that CE and FL do not enjoy robustness guarantees.

