# OpenReview forum: "A Curriculum Perspective of Robust Loss Functions"
_NeurIPS.cc/2022/Conference — NeurIPS 2022 Submitted_

### Official Review · Reviewer_racf · 2022-07-07

**Rating:** 6
**Confidence:** 4
**Soundness:** 3 good
**Presentation:** 3 good
**Contribution:** 3 good

**Summary:**

The paper provides a new understanding for two open questions in the learning with noisy labels literature: why robust loss functions can underfit the data distribution and why robust functions can be robust. The analyses are delivered by decoupling the equivalent sample weight from each loss function, which is further viewed as an implicit sample-weighting curriculum. Based on the proposed analytical framework, the authors experimentally studied the relationship between the underfitting of robust loss functions and the average effective learning rate (due to equivalent sample weights), and proposed solutions to mitigate underfitting. Besides, by putting the equivalent sample weights and training dynamics together, the authors showed the robust loss functions assign lower weights to wrong labels compared with CE.

**Questions:**

There are some minor questions:
1. The considered loss functions are relatively limited due to the fast-moving literature. There are several recent robust loss functions, such as peer loss and its extensions [R1-R2], Taylor cross-entropy [R3], label smoothing (positive or negative) [R4-R5]. It would be interesting to provide a library for the equivalent weights for these loss functions.
2. In addition to the loss functions that do not require estimating the noise transition matrix, is it possible to also find the equivalent weights for the loss correction/reweighting based methods, such as [40], [R6]?
3. It would also make the paper more convincing if Figure 4 could also be plotted for the above loss functions.
4. Current experiments only consider the synthetic label noise with simple noise generation regime. It is interesting to see whether the findings hold on real-world human annotations, e.g., CIFAR-N [R7].

[R1] Liu Y, Guo H. Peer loss functions: Learning from noisy labels without knowing noise rates. ICML 2020.

[R2] Cheng H, Zhu Z, Li X, Gong Y, Sun X, Liu Y. Learning with instance-dependent label noise: A sample sieve approach. ICLR 2021.

[R3] Feng L, Shu S, Lin Z, Lv F, Li L, An B. Can cross entropy loss be robust to label noise? IJCAI 2021.

[R4] Lukasik M, Bhojanapalli S, Menon A, Kumar S. Does label smoothing mitigate label noise? ICML 2020.

[R5] Wei J, Liu H, Liu T, Niu G, Sugiyama M, Liu Y. To Smooth or Not? When Label Smoothing Meets Noisy Labels. ICML 2022.

[R6] Liu T, Tao D. Classification with noisy labels by importance reweighting. IEEE TPAMI 2015.

[R7] Wei J, Zhu Z, Cheng H, Liu T, Niu G, Liu Y. Learning with noisy labels revisited: A study using real-world human annotations. ICLR 2022.

**Limitations:**

They discussed the limitations in Sec. 3.1 Potential negative societal impacts are not applicable.

**Strengths And Weaknesses:**

**Originality**

The idea and analytical framework are original and novel.

**Quality, clarity**

Both the theoretical part and the experiments are clear and of high-quality.

**Significance**

The reviewer believes this new finding could contribute sufficiently to the noisy learning literature, and guide the design of robust loss functions.

---

> ### Author Response · Authors · 2022-08-02
> **Extended Review of Loss Functions and Additional Results**
>
> Thanks for your insightful comments! We have revised our submission accordingly.
>
> Regarding Questions:
>
> 1. We have discussed them in the revised Apendix A.
>
>    1. Following your advice, we additionally made Appendix A an extended review for more loss functions and plots of their sample weighting functions with different hyperparameters, which can serve as a general reference. However, a thorough empirical investigation of all existing loss functions would make the main text less concise.
>
>     2.  Our derivations requires loss functions to be reduced to the "standard form" in Eq.5 (or Eq.7 in the revised version), i.e., a function of $\Delta_y$. Except for Taylor cross-entropy [R3], references [R1-5] additionally depends on $\Delta_i$, $i\neq y$, and thus does not fit this assumption. As shown in [R2], peer loss [R1] in its expectation essentially reduces to a cross entropy loss plus a confidence regularizer. Label smoothing and its generalized form [R5] is equivalent to cross entropy plus an entropy reducing [R4] or promoting [R5] regularizer. We dicuss these loss functions in Appendix A.4. However, examining how they interact with $\Delta_y$ distributions, though a very interesting question, can be out of scope for our submission, which focus on the implicit sample-weighting curriculums.
>
> 2. [40] and [R6] does not fit our analysis due to similar reasons as question 1. In [40], the corrected losses additionally depends on $\Delta_i$, $i\neq y$, which cannot reduce to an additional regularizer. In [R6], the weights for each sample are explicitly estimated for importance reweighting, rather than induced from a loss function. Incorporating all approaches on noisy robust training within an overarching framework is a fundamental quest. We believe that our summision can serve as a useful stepping stone.
>
> 3. In our revised version, we include results with CIFAR10N [R7] in Table 4 to improve the generality of our conclusions. In addition, in Fig. 8 of the revised Appendix F, we include similar plots as Fig.4 on CIFAR10N with more loss functions additionally reviewed in Appendix A1-A3. We carefully rescale the vertical axes to improve readability, and include the test accuracies for reference. To facilitate our analysis, we should know a priori which label is noisy, while avoiding complications like underfitting on CIFAR100. We thus narrow ourselves to CIFAR10 for Fig. 4 and Fig. 8.

---

> > ### Comment · Reviewer_racf · 2022-08-05
> > **Thank you for the response.**
> >
> > Thanks for the comprehensive analyses in Appendix A.
> >
> > I agree with your analyses on label smoothing and peer loss. Only defining sample weight wrt $\Delta_y$ may be a bit limited but I agree it may be out of the scope of the current paper.
> >
> > Table 4 with human noise is very interesting. It seems that GCE, MAE, AUL and AGCE could not work well under this noise. Could the author explain this observation? Is it possible to make the above loss functions better by tuning learning rate?

---

> > > ### Author Response · Authors · 2022-08-08
> > > **Thanks for you comments**
> > >
> > > Thanks for your comments!
> > >
> > > We tune for the best hyperparameter and learning rates for these losses. The default learning rate (0.01) achieves the best overall performance. The expected performance of AUL(68.43) and AGCE(68.94) can be improved to 72.69 and 74.53, respectively, with the best hyperparameters. MAE and GCE stays relatively the same. However, despite outperforming CE (62.49), they still fail to match the performance with the same level of symmetric label noise. We will keep the same results in Table 4 to avoid complications in hyperparameter settings.
> > >
> > > We hypothesize that the inferior performance can be due to the instance-dependent label noise of CIFAR10-N. In Fig. 1, instances of CIFAR10 are randomly emphasized at initialization, which can lead to a less representative expected gradient with instance-dependent label noise than with synthetic symmetric label noise, since the latter has more uniform noisy label-feature distribution. The skewed initial expected gradient may guide the optimizer to a worse parameter region, thus leads to worse performance.

---

### Official Review · Reviewer_CAua · 2022-07-09

**Rating:** 4
**Confidence:** 5
**Soundness:** 3 good
**Presentation:** 3 good
**Contribution:** 3 good

**Summary:**

In learning with noisy labels, designing certain loss functions is a promising direction to achieve noise robustness. This paper aims to use a new perspective (curriculum) to understand the key issues of robust loss functions. Typically, there are two issues in existing loss functions. 1). noise robustness, which is usually analyzed by symmetric condition; 2). underfitting problem, which is an open question to be explored.

Overall, this paper has following contributions:
1. This paper claims that the robust loss functions are implicitly weighted. Based on this finding, the authors explain the reason of underfitting problem (i.e. either due to small initialized weight or weight diminishing during training) and noise robustness (i.e., weights are different for clean and noisy samples).
2. This paper addresses underfitting problem by shifting or scaling the initialized weights.


**Questions:**

1. In Table 3, settting $a=2.6$ achieves optimal results for MAE. How to find this optimal hyperparameter? Does it depend on the complexity of dataset?

2. Does shitting or rescaling also work for other robust loss functions? such as NCE or AGCE?

3. Existing works explain the underfitting problem by gradient analysis (gradient coefficients are smoothed) or regularization theory (fitting ability is limited). How will you compare this work with other works?


**Limitations:**

Yes

**Strengths And Weaknesses:**

Strengths:
1. Using sample-weighting curriculum to understand underfitting problem is interesting.
2. The overall paper is well-written and makes sense.

Weaknesses or concerns:
1. Since this paper attempts to explain some open questions, no SOTA method is proposed. The overall novelty is limited.
2. The performance improvement of the proposed techniques (shifting and scaling) is marginal, especially at high noise rate. Compared to the existing methods (NCE+AUL or NCE+RCE), the performance of MAE+rescale does not outperform them much.
3. It seems that the proposed approaches are not practical and dependent on empirical observations. For example, it has to estimate the expectation of $\Delta_{y}$ in Eq. (9) based on several assumptions.  In Eq. (10), it seems that the hyperparameter $a$ also depends on complexity of data, which can be hard to find in practice. I am curious about the performance of MAR+rescale on real-world noisy datasets, such as Clothing1M and Webvision.
4. In Table 2, it shows that both AUL and AGCE suffer from severe underfitting problem. The paper only provides the results on MAE. Do shift or rescale work for AUL or AGCE? More experiment results should be conducted on these complex loss.
5. One of the proposed claim that "cross entropy can appear robust by adjusting learning rate" is not new idea. [1] has observed the similar results and adjusts the learning rate cyclically to achieve noise robustness.
6. In Eq. (5), the loss can be represented as a form of $w(\Delta_{y})$ and $\Delta_{y}$. This paper only considers the impact of $w(\Delta_{y})$ but does not further investigate the effect of $\Delta_{y}$ distribution (only provide the $\Delta_{y}$ distribution at at initialization in Figure 1). Will the $\Delta_{y}$ distribution change in training?

In summary, this paper proposes an interesting perspective to understanding key problems of noise robust loss functions. However, some claims are "hand-waving" to me and the theoretical analysis is incomplete. The experimental results are not fully supporting the advantage of the proposed methods when compared with recent noisy-label learning methods. I hope that the authors can clarify these points in the rebuttal.

[1] Huang, Jinchi, et al. "O2u-net: A simple noisy label detection approach for deep neural networks." Proceedings of the IEEE/CVF international conference on computer vision. 2019.

---

> ### Author Response · Authors · 2022-08-02
> **Clarifying Misunderstandings and Additional Results**
>
> Thanks for your thoughtful comments! We have revised our submission accordingly and include the reference.
>
> Regarding the Weakness:
>
> 1. As mentioned in the general comment, we do not aim for "yet another SOTA method". Our focus lies in providing empirical evidence for our novel curriculum perspective, which can help design and analyze robust loss functions, a field previously dominated by theoretical bounds to the risk minimizers. We have clarified this point to avoid misconceptions in our revision.
> 2. When addressing underfitting, we do not claim that beating the SOTA is our major contribution. The baseline should be the vanilla MAE that heavily underfits; SOTA results  are included just as context for comparison. The inferior performance at high noise rate lies in a trade-off between underfitting and noise robustness of our simple fix, which we briefly discuss in line 217-220 in our revision.  Our curriculum perspective can lead to more performant approaches, such as explicitly designing the sample-weighting curriculums, which we leave for future work.
> 3. The derived distribution of $\Delta_y$ can approximate standard settings as we demonstrate in Fig.6 of Appendix E in the origional submission. In Fig. 7 from Appendix E in our revised version, we compare our simulations to additional settings of WebVision. The hyperparameter $a$, like any hyperparameters such as learning rate, are dataset dependent and should be tuned on a validation set. We include results on Webvision with different number of classes in Table 11 of Appendix E in the revised version. Scaling improves MAE on WebVision50 (the standard "mini" setting [1-2]) from 3.68 to 66.72, comapring to 66.40 of CE.  Results on WebVision 400 have not yet converge, we will update it in later versions.
> 4. With limited computation budget, we opt for loss functions that severely underfits as typical example. In our revised version we include results of AGCE using $a=4.5$ in Table 3,9,10. Without label noise, scaling improves AGCE from 49.27 to 67.20. With symmetric label noise and $\eta = 0.4$, AGCE improves from 47.76 to 56.32.
> 5. This is a minor point and we only emphaze the unexpectiveness when viewed from existing robustness conditions (line 43). In addition, the listed reference uses the cyclic learning schedule only to collect statistics for noisy label detection. To achive noise robustness they drop the detected noisy samples and retrain the model. In contrast, our results shows that simply changing the learning rate can by itself achieve noise robustness. We include the reference when discussing curriculum based approaches (reference [25] in line 33 of the revised version)
> 6. With our curriculum perspective, loss functions we examined only differs in their weights $w(\Delta_y)$. Thus we examine their interaction with different training dynamics, which manifest themselves as initialization and changes in $\Delta_y$ distribution. The change of $\Delta_y$ distribution and its relation to noise robustness is the main topic of section 4.3. As shown in Fig.4, it changes in an interesting way.
>
> Regarding the Questions:
>
> 1. See response for the Weakness 3. $a$ depends on dataset and loss functions and is selected on the validation set.
> 2. It does not work for NCE as NCE underfits due to fast diminising sample weights (line 171-176, section 4.1). We restate this point in line 194-195 of the revised version. It works on AGCE as discussed in response for Weakness 4.
> 3. For detailed comparisons, could you please share some references? Besides, we do not aim for a generic discussion on underfitting, but only focus on a possible explanation on the underfitting issue of selected robust loss functions. The effectiveness of our simple fix empirically supports our explanation.
>
> Reference
> 1. Normalized loss functions for deep learning with noisy labels. [ICML2020]
> 2. Asymmetric loss functions for learning with noisy labels. [ICML2021]

---

### Official Review · Reviewer_xcSK · 2022-07-10

**Rating:** 5
**Confidence:** 2
**Soundness:** 2 fair
**Presentation:** 3 good
**Contribution:** 3 good

**Summary:**

This paper offers a curriculum perspective in analyzing robust loss functions, which argues that each loss function implicitly induces a sample weighting curriculum. This analysis reveals that underfitting of robust losses can be explained by marginal effective learning rate and marginal initial sample weight. The paper proposes to address underfitting by adapting the sample-weighting curriculums.

**Questions:**

Please refer to my weakness section.

**Strengths And Weaknesses:**

Strengthes:

1. The paper is well-motivated from the existing problems of robust loss functions.
2. The analysis on the sample weighting curriculum offers an insightful perspective on the underfitting effect of robust loss functions.
3. Motivated by the sample weighting analysis, the paper proposes sample-weighting curriculums that can address the underfitting issue.

Weaknesses:
My main concern is that it is unclear how general the findings would hold. The experiments are conducted in a posthoc manner where it is first known that CIFAR100 suffer from the underfitting issue. Given that CIFAR100 is the only dataset with underfitting, it is unclear whether the findings can generalize to other datasets with similar issues. How to ensure that the sample weight curriculum is not a spurious correlation imposed by the CIFAR100 dataset?

---

> ### Author Response · Authors · 2022-08-02
> **Clarifications and Results on Broader Settings**
>
> Thanks for your thoughtful comments!
>
> Regarding the Weakness:
>
> - The fact that robust losses can underfit on difficult task is well backed by the literature (reference [1, 10, 12, 13] in both the original and revised version), where CIFAR100 is regarded as a typical example. Our work is based on these established results.
>   - In our revised version, we include results on WebVision [1] with different number of classes (10, 50, 200, 400) in Table 11 of Appendix E. The vanilla MAE does underfit. Scaling improves MAE on WebVision50 (the standard mini setting) from 3.68 to 66.72, comapring to 66.40 of CE. Results on WebVision 400 have not yet converge, we will update it in later versions. Similar results can be find in Table 3 with human noisy labels on CIFAR100N [2].
> - Sample weighting curriculums with weighting function $w(\Delta_y)$ are not imposed by specific datasets but induced by loss functions we reviewed in section 2.2. Please refer to our derivations in section 3. The derivation to reach the sample-weighting curriculum of each loss functions is our major contribution.
>
> Reference
> 1. WebVision Database: Visual Learning and Understanding from Web Data. [arxiv 2017]
> 2. Learning with Noisy Labels Revisited: A Study Using Real-World Human Annotations. [ICLR 2022]

---

> > ### Comment · Reviewer_xcSK · 2022-08-08
> > **Thanks for the response**
> >
> > I'd like to thank the authors for their response and results on the WebVision dataset. All my questions have been answered.

---

### Official Review · Reviewer_a8RC · 2022-07-11

**Rating:** 6
**Confidence:** 4
**Soundness:** 3 good
**Presentation:** 2 fair
**Contribution:** 2 fair

**Summary:**

When learning with noisy labels under certain robust loss designs, the authors focus on addressing two open questions: (1) Why do robust losses underfit? and (2) What affects the robustness of a loss? The two questions are explained through the view of sample-weighting curriculum perspective for certain robust losses. To overcome the underfitting issue, the authors propose to adapt the sample-weighting curriculums. To make loss more robust, the authors further show that adjusting the learning rate schedule helps.

**Questions:**

My concerns/questions/suggestions are listed below:

1. In Line 62, a pointer/reference to the argument "With $\eta<(k-1)<k, L$ is robust against symmetric label noise" would be appreciated.

2. In Line 64, $\ell$ is never defined, also for $\Delta p$ in Eqn. (2).

3. Typo: Line 108 "a sampled is learnt".

4. In Table 3, although it seems that MAE has pretty bad performances on the mentioned tasks, would fine-tuning influence much on the performance of MAE? Since 1.00 accuracy on CIFAR-100 is merely random guessing or predicting all samples to be the same class.

**Limitations:**

See Weakness 2.

**Strengths And Weaknesses:**

$\textsf{Strengths:}$
1. This work connects several popular robust loss designs to sample-weighting curriculum.

2. Empirically, the authors explain the two open questions in the literature of learning-with-noisy-labels. The introduce of marginal effective learning rate looks interesting and helps with explaining the underfitting issue. And the shifting of soft-margin mitigates the underfitting, especially when the number of classes is large.

$\textsf{Weaknesses:}$

1. Presentation could be further improved, i.e., notation definitions, and argument statements  (please refer to Section "Questions" for more details.)

2. A potential weakness of this work is that all the analysis focuses on the class-dependent label noise, although it enables easiness of theoretical analysis, it was shown that real-world label noise tends to be instance-dependent [1]. And the majority section of this work falls into empirical explorations. It would be interesting to see whether the conclusions still hold for real-world noisy data, either small-scaled ones (CIFAR-10N, CIFAR-100N real-world noisy CIFAR [2]) or the large-scale Clothing 1M dataset.

$\textsf{References:}$

1. Part-dependent Label Noise: Towards Instance-dependent Label Noise. [NeurIPS 2020]

2. Learning with Noisy Labels Revisited: A Study Using Real-World Human Annotations. [ICLR 2022]

3. Learning From Massive Noisy Labeled Data for Image Classification. [CVPR 2015]

---

> ### Author Response · Authors · 2022-08-02
> **Improved Presentation and Results on Broader Settings**
>
> Thanks for your constructive comments! We have revised our submission accordingly.
>
> Regarding Weakness:
>
> 1. We have carefully revised our submission: we remove unnecessary symbols and describe the meaning and intuitions for each symbol and formula. Section 2.1, 3.1 and 4.2 in the revised version should be much easier to follow.
>
> 2. In our revised version:
>     1. In section 4.2 where we test our fix for the underfitting issue, we add results on CIFAR100-N   [1] in Table 3. We also subsample 10, 50, 200, 400 classes from the ImageNet-size noisy dataset WebVision [2] and report results in Table 11 of Appendix E, which can cover a broad array of real settings. Results on WebVision 400 have not yet converge, we will update it in later versions. Due to limited time and computation budget, we will consider adding Clothing1M in later revisions.
>     2. In section 4.3 where we investigate noise robustness, we add results on CIFAR10-N in Table 4. We also add similar plots as Fig.4 for a broad array of loss functions with CIFAR10-N in Fig. 8 of Appendix F. We omit CIFAR100-N as it involves complications of underfitting. Since analysis in Table 4 and Figure 4 requires knowing whether a label is noisy, real world noisy datasets like WebVision are not applicable here.
>
> Regarding Questions:
>
> 1. The reference is at line 61. We have addressed it at line 67 in our revision
> 2. We define symbols and its intuition at line 68 and 70 in our revision
> 3. Have changed it to "a sample is learnt." at line 115 in our revision
> 4. We try fintuning a pre-trained ResNet34 on clean CIFAR100. We keep the same settings and tune the learning rate for the best hyperparameters. With learning rate 0.01, results are 58.84 for CE and 16.92 for MAE . Scaling MAE by $a=2.6$ improves it to 59.15. Finetuning mitigates underfitting of MAE to some extent, but the performance gap compared to CE is still not negligible.
>
> Reference
>
> 1. Learning with Noisy Labels Revisited: A Study Using Real-World Human Annotations. [ICLR 2022]
> 2. WebVision Database: Visual Learning and Understanding from Web Data. [arxiv 2017]

---

### Author Response · Authors · 2022-08-02
**A Clarification of Our Contribution**

Thanks for the constructive and insightful comments from all reviewers!

Our major contribution is the curriculum perspective for robust loss functions. However, the significance of such perspective, as recognized by Reviewer racf, may be a little bit underrated:

1. Instead of presenting "yet another SOTA method", we focus on illustrating the intuitions from the novel curriculum perspective and how they matches empirical results, which can lead to better design of robsut loss functions and curriculums. In our main text, we have also emphasized the simplicity of our fix to the underfitting issue. We think that novelty can be more than SOTA results.
2. The dominant paradigm on robust loss functions focuses on bounding the difference between risk minimizers on noisy&clean data, which is agnostic to training dynamics (line 60-61) to reach them. Such framework cannot fully explain the two questions we address in this work (line 21-22) and results we discover (line 43-44). With this context, our work unifies a series of robust loss functions under the novel curriculum perspective, which helps easily analyze their interaction with different training dynamics and resolve the aforemention questions.

We have made these points more apparent in our revision.

---

### Meta-Review · Area_Chair_oY49 · 2022-08-24

**Recommendation:** Reject
**Confidence:** Certain

**Metareview:**

The paper focuses on investigating the underfitting issue of certain robust loss functions and why functions deviating from theoretical robustness conditions can still be robust. A key vehicle to support the analysis is a standard form with equivalent gradients that enables the investigation of different robust loss functions from a curriculum learning perspective. In this standard form, each loss function is factorized into a sample weight and an implicit loss. Empirical studies have been conducted to show the interaction between sample-weighing curricula and different training dynamics to gain deeper insights on the underfitting issue and why loss combination can mitigate this. A shifting/rescaling strategy is also developed to adapt the sample-weighing curricula so that the underfitting issue can be addressed.

The paper contributes to an important area in machine learning given the labeling errors commonly exist in most real-world datasets. The proposed work complements most existing research that focuses on the design of robust loss functions and bounding the risk minimizer over these functions. The derivation of the standard form with equivalent gradients is new, which nicely connects a set of robust loss functions with a sample-weighting curriculum. The analysis afterwards also shows some interesting insights on the open questions being considered.

Major concerns from reviewers include whether some of the important findings will hold in general. These concerns are related to the evaluation on limited datasets (e.g., only CIFAR100 suffers from underfitting) and only MAE is tested for the effectiveness of the shifting/rescaling strategy. While the authors’ rebuttal addresses some of these critical comments, a more comprehensive evaluation may still be needed to make a convincing case. Furthermore, while the analysis is interesting, in order to justify it is truly useful, it is still necessary to show how it helps to inform the design of better robust loss functions or lead to a better learning process that can further improve the current models. The proposed shifting/rescaling strategy shows some promise, but it is designed in a rather heuristic way. It heavily relies on a critical hyper-parameter that is dataset dependent and may be quite sensitive and thus difficult to be set properly in practice.

The Senior AC and AC discussed the paper and the authors concerns and feel that the paper is not yet above the NeurIPS publication bar. The authors are encouraged to take the reviewer feedback into account to prepare an improved draft which they can submit to an upcoming conference.


**Award:**

No

---

### Decision · Program_Chairs · 2022-09-14

Reject